# Information Structures for Causally Explainable Decisions

**DOI:** 10.3390/e23050601

**Published:** 2021-05-13

**Authors:** Louis Anthony Cox

**Affiliations:** Department of Business Analytics, University of Colorado School of Business, and MoirAI, 503 N. Franklin Street, Denver, CO 80218, USA; tony.coxjr@ucdenver.edu

**Keywords:** explainable AI, XAI, causality, decision analysis, information, explanation, Bayesian networks, reinforcement learning, partially observable Markov decision processes, stochastic control

## Abstract

For an AI agent to make trustworthy decision recommendations under uncertainty on behalf of human principals, it should be able to explain *why* its recommended decisions make preferred outcomes more likely and what risks they entail. Such rationales use causal models to link potential courses of action to resulting outcome probabilities. They reflect an understanding of possible actions, preferred outcomes, the effects of action on outcome probabilities, and acceptable risks and trade-offs—the standard ingredients of normative theories of decision-making under uncertainty, such as expected utility theory. Competent AI advisory systems should also notice changes that might affect a user’s plans and goals. In response, they should apply both learned patterns for quick response (analogous to fast, intuitive “System 1” decision-making in human psychology) and also slower causal inference and simulation, decision optimization, and planning algorithms (analogous to deliberative “System 2” decision-making in human psychology) to decide how best to respond to changing conditions. Concepts of conditional independence, conditional probability tables (CPTs) or models, causality, heuristic search for optimal plans, uncertainty reduction, and value of information (VoI) provide a rich, principled framework for recognizing and responding to relevant changes and features of decision problems via both learned and calculated responses. This paper reviews how these and related concepts can be used to identify probabilistic causal dependencies among variables, detect changes that matter for achieving goals, represent them efficiently to support responses on multiple time scales, and evaluate and update causal models and plans in light of new data. The resulting causally explainable decisions make efficient use of available information to achieve goals in uncertain environments.

## 1. Introduction: Creating More Trustworthy AI/ML for Acting under Risk and Uncertainty

How can the predictions, decisions and recommendations made by artificial intelligence and machine learning (AI/ML) systems be made more trustworthy, transparent, and intelligible? Enabling AI/ML systems to explain the reasons for their recommendations in terms that make sense to humans would surely help. This paper reviews a set of concepts, principles and methods for creating AI/ML systems that recommend (or execute, for autonomous systems) appropriate actions—those we would want them to take, or at least understand and approve of their rationales for taking, if they were acting on our behalf. The key ideas are based on *causal models* of the relationship between actions and outcome probabilities. To a limited but useful extent, current causal models enable appropriate decisions even under unforeseen conditions and in response to new and unanticipated events. Such “causal artificial intelligence” (CAI) principles might also be useful in improving and explaining decision and policy recommendations in human organizations when risk, uncertainty, and novelty make the consequences of different courses of action hard to predict and, thus, make collecting information for predicting them valuable [1].

CAI builds on the intuition that systems that can explain the rationales for their inferences, predictions, recommendations, and behaviors in clear cause-and-effect terms are likely to be more trusted (and, perhaps, more trustworthy) than those that cannot. It applies principles of information theory and closely related probabilistic and statistical concepts of conditional independence, probabilistic dependence (e.g., conditional probabilities), causality, uncertainty reduction, and value of information (VoI) to model probabilistic dependencies among variables and to infer probable consequences caused by alternative courses of action. To recommend best decisions despite incomplete causal knowledge and information, CAI seeks not only to identify facts that make current observations less surprising, and in this sense to explain them; but also to identify actions, policies, and plans that make preferred future outcomes more probable, and in this sense, explain how to achieve them. CAI applies statistical tests for conditional independence (or, conversely, mutual information) and other relationships (e.g., directed information flows and invariant causal prediction properties, reviewed later) among random variables to identify causal regularities that are consistent with multiple datasets, thereby enabling generalization from experience and prediction of probable consequences of courses of action in new settings [2]. Such causal generalization may be essential for acting effectively, as well as for explaining the basis for actions or recommendations, when confronting novel situations and unanticipated risks.

Current AI systems are typically most fragile and least trustworthy in novel situations because they lack common-sense knowledge and the ability to reason effectively (and causally) about likely consequences of actions when relevant prior data with stable predictive patterns are not available. CAI seeks to help bridge this crucial gap between experiential learning and the need to act effectively under novel conditions by applying causal generalizations from past observations. It recognizes the tight link between *causally effective plans*, meaning plans (i.e., sequences of actions contingent on events) that make preferred outcomes more probable, and *causal explanations* for preferred courses of action. Both exploit the fact that causes provide unique information about their direct effects (or joint distributions of effects): conditioning on levels of other variables does not remove the statistical dependency of effects on their direct causes. Causes and their direct effects have positive mutual information, and this information can be used to identify courses of action that make preferred outcomes more likely, hence less surprising. Intuitively, causally effective decision-making can be thought of as mapping *observations*—signals received by an agent from its environment—to *decisions* and resulting behaviors that are calculated to change outcome probabilities to make preferred outcomes more likely. Decisions are implemented by control signals—typically transduced by effectors which may be unreliable or slow—sent into the agent’s environment to change outcome probabilities. A completely specified probabilistic causal model predicts conditional probabilities of outcomes, given observations and actions. This provides the information needed to optimize actions. Even partial information about causal dependencies among variables can help to decide what additional information to seek next to formulate more causally effective policies. To act effectively, an agent must receive and process observations and transmit control signals quickly enough to keep up with changes in its environment.

Figure 1 summarizes key concepts and methods reviewed in subsequent sections and shows how they fit together. *Observations* (upper left) provide information about the underlying state (lower left) of the system or environment via an *information channel*, i.e., a probabilistic mapping from the state to observations. *Actions* (lower right) cause state transitions and associated costs or benefits, generically referred to as rewards (bottom) via a *causal model*, i.e., a probabilistic mapping from current state-action pairs to conditional probabilities of next-state and reward pairs. Table 1 lists several specific classes of causal models, discussed later, in rough order of increasing generality and flexibility in representing uncertainty. Actions are selected by *policies*, also called decision rules, strategies, or control laws (upper right), which, at their most general, are probabilistic mappings from observations to control signals sent to actuators; these control signals are the decision-maker or controller’s choice. The mapping from choices to actions may be probabilistic if actuators are not entirely reliable; in this case, the capacity of the control channel and actuators to transfer information from control signals to future states of the system limits the possibilities for control. In many settings, actions are implemented via hierarchies of learned skills (i.e., abilities to complete tasks and subtasks); what can be done in a situation depends, in part, on the repertoire of skills that have been acquired [3]. In Figure 2, *observations* are explained by underlying states (and the information channels via which they are observed). By contrast, rational *decisions* are explained via optimization of decision-rules (i.e., policies). If knowledge of the causal model and the other components in Figure 1 is inadequate to support full optimization, then reinforcement learning or other adaptive control methods, together with optimization heuristics, are used to improve policies over time. The vast majority of “explainable AI” (XAI) research to date has focused on explaining observations, as in diagnostic systems, predictions, and prediction-driven recommendations [4]. Such explanations emphasize the left side of Figure 1, where observations are used to draw inferences about states, which can then be used to predict further observations. A principal goal of this paper is to help extend XAI to more fully explain the rationales for recommended decisions and policies. Such explanations draw also on the right side of Figure 1, using preferences for outcomes (i.e., rewards), choice sets (e.g., possible control signals), causal models, and optimization of policies as key explanatory constructs, in addition to observations and inferences.

**Figure 1 entropy-23-00601-f001:**
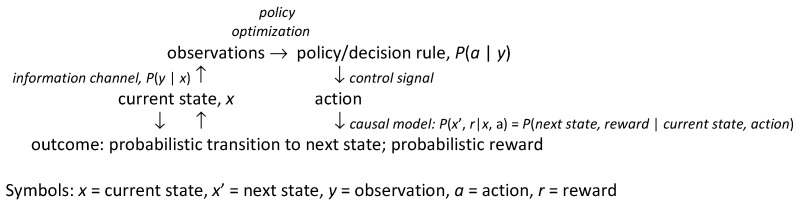
Summary of key ideas and methods used in causal AI (CAI) to explain decisions.

The following sections seek to show how these intuitions can be sharpened and formalized and how they can be implemented in computationally effective CAI algorithms to support and explain decision recommendations for several widely applied classes of probabilistic causal models. One goal is to review how current causal AI/ML methods support and explain causally effective decisions—decisions that make preferred outcomes more probable—in practical applications, such as allocating a budget across advertising channels that interact in affecting consumer preferences and sales. We seek to present an accessible exposition, review and synthesis of CAI ideas and advances from several decades of AI/ML progress for a broad audience that might include decision analysts, risk analysts, decision science researchers, psychologists, and policy analysts, as well as AI/ML researchers. A second goal is to propose a framework clarifying the types of information and argument structure needed to provide convincing and responsible causal explanations for CAI-based decision and policy recommendations.

**Table 1 entropy-23-00601-t001:** Some important probabilistic causal models.

Probabilistic Causal Models, in Order of Increasing Generality	Knowledge Representation, Inference, and Decision Optimization Assumptions
Decision trees [5], event trees (decision trees without decisions); fault trees, event trees, bow tie diagrams [6]	Event trees show possible sequences of events (realizations of random variables). Decision trees are event trees augmented with choice nodes and utilities at the leaves of the tree. Fault trees are trees of binary logical events and deterministic logic gates, supporting bottom-up inference from low-level events to top-level events (e.g., systems failure). Bow-tie diagrams integrate fault trees leading up to an event and event trees following it.
Bayesian networks (BNs), dynamic BNs, causal BNs Influence diagrams (IDs) are BNs with decision nodes and utility nodes [7]	Random variables (nodes) are linked by probabilistic dependencies (described by conditional probability tables, CPTs). In a DBN, variables can change over time. In a causal BN, changing a variable changes the probability distributions of its children in a directed acyclic graph (DAG). Bayesian inference of unobserved quantities from observed (or assumed) ones can proceed in any direction.
Markov decision process (MDP) optimization models, can be risk-sensitive	Markov transition assumptions, observed states, actions completed without delays.
Partially observable MDPS (POMDPs)	States are not directly observed, but must be inferred from observations (signals, symptoms, data) via information channels, *P*(*observation* = *y*|*state* = *x*)
PO semi-MDPs (POSMDPs); behavior trees	Actions take random amounts of time to complete and may fail.
Discrete-event simulation models	Realistic lags and dependencies among events are modeled by state-dependent conditional intensities for individual-level transitions.
Causal simulation-optimization models	Known models. Inference and optimization can be NP-hard and may require heuristics, such as Monte Carlo Tree Search (MCTS).
Model ensemble optimization; reinforcement learning (RL) with initially unknown or uncertain causal models	Unknown/uncertain models. Probabilistic causal relationships between actions and consequences (e.g., rewards and state transitions) are learned via (heuristic-guided) trial and error.

## 2. Methodology and Applications

To accomplish this review, synthesis, and exposition of CAI for explaining decisions, we first consider the traditional concept of explanation in statistics as the proportion of variance in a dependent variable in a regression model that is “explained” by differences in the independent variables on which it depends. A different concept is needed to explain decisions, since regression coefficients only address how to predict dependent variables from observed independent variables, but not the causal question of how intervening to exogenously change independent variables would change probability distributions of dependent variables [8]. We therefore examine the structure of causal explanations for decisions in the causal models in Table 1. These models have been widely used in causal analytics, risk analysis, and applied AI/ML to represent probabilistic dependencies of outcomes (e.g., rewards and next states in a Markov decision process) on actions [6]. For each model, we discuss how the concepts in Figure 1 can be used to recommend decisions and explain their rationales. The discussion includes relatively recent developments (e.g., integration of Thompson sampling into Monte Carlo Tree Search) while noting foundational works in prescriptive decision analysis and decision optimization (e.g., [7,9,10]).

Our focus is on CAI concepts, principles and methods for causal explanations of decisions, but a variety of practical applications have been described in industrial engineering and industrial control, managerial economics (e.g., for forestry or fishery management), personalized medicine, supply chain management, logistics optimization, urban traffic control, robotics, autonomous vehicle and drone control, pest management in ecosystems, financial investments, game-playing, and other areas of applied risk analysis (e.g., [6]). Many of these applications have focused solely on decision optimization, rather than also on decision explanation, under risk and uncertainty. This motivates our focus on the structure of causal explanations under risk and uncertainty. However, to understand how CAI methods and explanations can improve the trustworthiness of causal inferences and intervention decision recommendations in practice, we recommend recent analyses and applications of computational causal methods in Systems Biology for cancer research. Although current CAI methods do not yet fully automate valid causal discovery with high reliability [11], causal discovery and understanding of low-level (molecular-biological) pathways are increasingly able to inform, and build confidence in, high-level public health policies by helping target the right causal factors at the macro-level (e.g., diet, exposures) to be causally effective in reducing risks [12]. Explaining how interventions cause desired changes helps to select effective interventions.

## 3. The Structure of Traditional Statistical Explanations

What should explainable AI (XAI) explain? Observations (why did an observed event happen?), predictions (what might happen next, and how probable is it?), and recommendations (what should we do now to make preferred outcomes more probable?) are natural candidates for diagnostic, predictive, and decision support (prescriptive) systems, respectively. Explaining actions, choices, and behaviors by AIs becomes more crucial as increasingly autonomous AI-directed systems interact more with people. Most of this paper focuses on explaining *decision recommendations*: if an AI system recommends an action or policy, how can and should it explain and justify its recommendation? We first contrast this with explaining *observations*, the more usual focus of descriptive, diagnostic, predictive, and prescriptive systems that rely on stable patterns in data to explain some proportion of the variation in observed outcomes across cases.

It is natural in many cases to regard a system under study as having inputs, some of which may be controllable, and outputs, some of which are observed. Learning to explain observed output values in terms of combinations of input values that make them expected, or at least less surprising, is a fundamental task for statistics and machine learning. Statistical models posit a simple, powerful explanatory structure of the following form [13]:output = f(inputs,noise)(1)

Statistical regression models simplify this further, often to the following form:expected value of output = f(inputs) + noise(2)

The expected value of a dependent variable (the “output” of the regression model—a quantity, response, or outcome whose value is to be predicted or explained) is expressed as a deterministic function of variables on which it depends (the “inputs” or independent variables). The actual value of the dependent variable for each case is modeled as a random variable, typically (but not necessarily) the sum of the predicted value and a random noise or error term having zero mean. The regression approach thus postulates that variability in observed output values is “explained” as the sum of predicted values and unpredictable error or noise. More generally, the mean of the conditional probability distribution of a dependent variable, given the values of the variables on which it directly depends (its parents in a directed acyclic graph (DAG) or probabilistic graphical model of dependencies among variables) constitutes a *response surface model* (RSM) for the dependent variable (output) as a function of the independent variables (inputs) (see Figure 8 for an example).

As an explanation for output values, a statistical model does not run deep. It does not reveal *why* the expected value of the output depends on the inputs, or *how* (or whether) changes in the inputs propagate along causal pathways of successive mechanisms to change the output so that the regression model is again satisfied. Such deeper explanations are the province of causal modeling. Statistical regression models merely describe observed patterns and quantify the proportion of variance in outputs that is “explained” by differences in inputs, given the model assumptions. Figure 2 shows an example. In a dataset recording heights of parents and their adult offspring, we might wonder how much of the variance in offspring heights (the *Height* variable on the vertical axis in Figure 2) is explained by differences in the father’s height (*Father*, on the horizontal axis).

**Figure 2 entropy-23-00601-f002:**
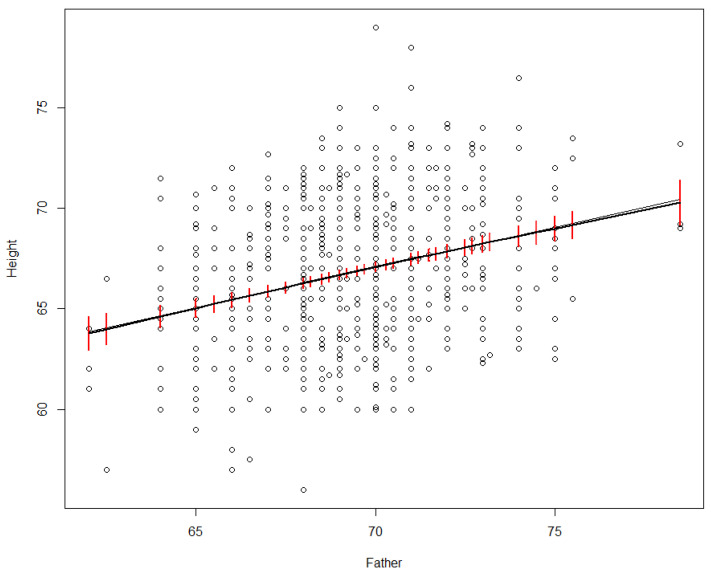
Linear (black line) and nonparametric smoothing (grey line) regression models for adult offspring’s height vs. father’s height in inches. 95% confidence intervals for the smoothing regrssion are in red.


# Data Source and R code for Figure 2
#Get dataset from web site, rename it as dataframe "df", attach it to current session
df <- read.csv(url("https://raw.githubusercontent.com/dspiegel29/ArtofStatistics/master/05-1-sons-fathers-heights/05-1-galton-x.csv")); attach(df);
# Fit and plot simple linear regression model
plot(Father, Height); lines(Father, predict(lm(Height~Father)));
# Fit and plot nonparametric regression model
library(npreg); smod<- sm(Height~Father); plotci(Father, smod$fitted.values, smod$se.fit, col = "red", bars = TRUE, xlab = "Height", ylab = "Father", col = "red", add=TRUE)

A simple linear regression model, represented by the black line in Figure 2, shows that the proportion of variance in *Height* explained by *Father* is about 7.5% [the adjusted R-squared value, produced in R via summary(lm(Height~Father, data = df))], corresponding to the fit of the black line to the scatterplot in Figure 2). Nonparametric (smoothing) regression produces an almost identical line and R^2^ value (it is shown by the smooth grey curve in Figure 2, which falls almost on top of the black line), but now the straight-line relation between *Father* and *Height* is a discovery rather than an assumption, as smoothing regression allows many other shapes. Various nonparametric smoothing (e.g., loess, kernel density, spline) and parametric (e.g., polynomial) regression models give closely similar fits; we relegate details to the underlying software packages, such as package *npreg* in R, which was used to produce the nonparametric regression curve in Figure 2. The key point is just that differences in the father’s height “explain” (in the statistical sense) a small but significant (i.e., non-zero with high statistical confidence) proportion of the observed variance in *Height* across offspring. Even such a partial explanation has practical value for prediction: it shows that the father’s height helps to predict offspring height. However, it does not address causal questions, such as how or whether the distribution of *Height* in a population would change if the distribution of *Father* heights were to change. For example, if it turned out that tall men tend to marry tall women, and the mother’s height alone affects offspring height, then the observed statistical association between *Father* and *Height* might be entirely explained by the mediating variable of the mother’s height. In this sense, the statistical “explanation” of variance in *Height* by differences in *Father* has nothing necessarily to do with causal mechanisms or explanations.

Including more independent variables, also called “explanatory variables,” on the right side of a regression model can increase the proportion of explained variance in the dependent variable. For example, including mother’s height increases the proportion of explained variance in offspring heights from 7.5% to 10.7% (adjusted R^2^ = 0.1069, as shown by the R commands attach(df); model <− lm(Height~Father + Mother); summary(model)); including offspring’s sex increases the R^2^ to 0.6385 (using summary(lm(Height~Father + Mother + Gender)). Figure 3 shows response surface models (RSMs) for this dataset, illustrating that offspring *Height* increases more steeply with *Father* height than with *Mother* height, and that *Gender* plays a larger role than either and does not interact with parent heights, insofar as the two sheets for men and women are approximately parallel.

**Figure 3 entropy-23-00601-f003:**
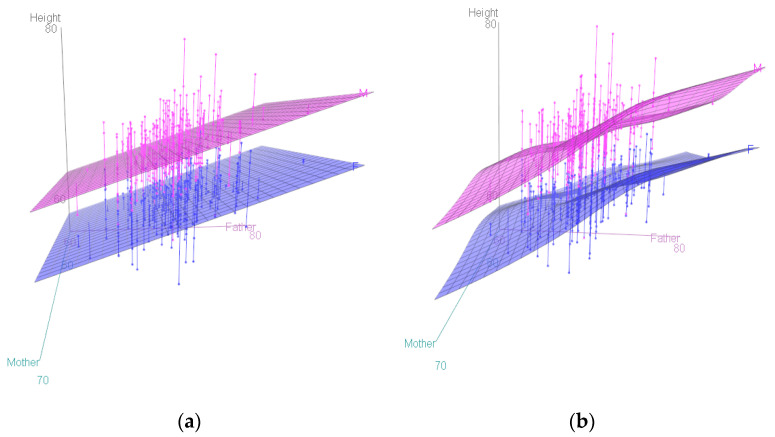
Linear (**a**) and smooth nonparametric (**b**) response surface models (RSMs) and data points for offspring heights.


#R code for Figure 3: library(car); scatter3d(Height~Father + Mother | as.factor(Gender), fit = c("linear" ));scatter3d(Height~Father + Mother | as.factor(Gender), fit = c("smooth"))

## 4. The Structure of Explanations in Causal Bayesian Networks (BNs)

Bayesian network (BN) models and more general probabilistic graphical models (e.g., with some undirected edges) generalize RSMs by quantifying probabilistic dependencies among all of the variables in a model, with no need for just one dependent variable. In a BN, nodes represent variables, and arrows between nodes represent statistical dependence relationships between variables. The response surface concept is generalized to that of a *conditional probability table* (CPT); we use this as a generic term to denote any probability model (not necessarily a table) specifying the *conditional probability distribution* of a random variable, given the values of its parents. (Nodes with no parents, e.g., input nodes with only outward-directed arrows, have marginal (unconditional) probability distributions, but we subsume these as special cases of CPTs). A fully specified BN is a DAG with a CPT at each node. As in regression models and RSMs, part of the variation in a dependent variable is “explained” by differences in the values of other variables on which it depends, i.e., variables in its CPT.

### 4.1. Explaining Direct, Indirect (Mediated), and Total Effects

Figure 4 shows a probabilistic graphical model with its arrows indicating that the value of *Height* depends on *Father*, *Mother*, and *Gender*. More specifically, the arrows signify that *Height* is not conditionally independent of any of the variables to which it is connected, even after conditioning on the other two; thus, for example, the hypothesis that the statistical effect of *Father* on *Height* is fully explained (or explained away) by *Mother* as a mediating variable is rejected in this model (since otherwise *Height* would be conditionally independent of *Father* given *Mother*, and there would be no direct link between *Father* and *Height*). In general, two nodes (variables) are connected in such a probabilistic graphical model if, and only if, they are found to be informative about each other, i.e., the null hypothesis of conditional independence between them is rejected, implying that they have positive mutual information even after conditioning on other variables. Such graphs enable a slightly deeper level of causal explanation for observations than do regression models, by distinguishing between, and quantifying, direct and indirect (mediated) causation. If a person is observed to be surprisingly tall or short, a partial explanation may be given in terms of the parents’ heights: the observed height may be less surprising, or more probable, after conditioning on relevant information about (a) the parents’ heights (explanatory inputs) and (b) the CPT for *Height* (i.e., knowledge of how inputs affect the conditional distribution of the *Height* output). This is similar to regression model-based explanations, with the CPT playing the role of the regression model and the parents of the outcome being the explanatory variables. However, the probabilistic graphical model goes beyond regression modeling by distinguishing between direct and indirect causes: a person’s unusual height might be partially explained by the direct effect of his or her father’s height, but also by the direct effect of his or her mother’s height, which in turn may be correlated with, and in that sense partly explained by, the father’s height. Thus, the *total effect* of the father’s height on the expected value (or conditional probability distribution) of the offspring’s height consists of both a *direct effect* and an *indirect effect* mediated by the mother’s height [14].

**Figure 4 entropy-23-00601-f004:**
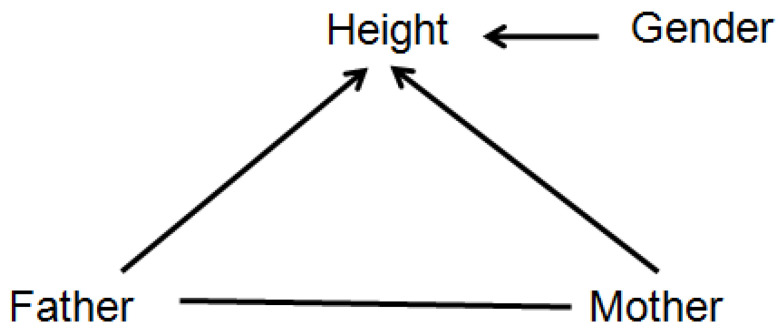
A probabilistic graphical model for the *Height* dataset.

In more detail, to be well-defined, counterfactual causal questions about how offspring height probabilities would differ if the father’s height was different must specify whether the mother’s height should be held fixed—and if so, at what level. Holding the mother’s height fixed in considering how the offspring height distribution varies with the father’s height isolates the *controlled direct effect* (or “partial” effect) of the father’s height on the conditional probability distribution of the offspring height [14]. Alternatively, if the mother’s height is allowed to vary realistically based on the father’s height, then we quantify the *total* (direct plus indirect, mediated by the mother’s height) effect of the father’s height. Distinctions among direct (or partial), indirect, and total effects of one variable on another, while holding some other variables fixed at specified levels, are causal concepts with no counterparts in statistical regression, as the concept of “*holding fixed*” is not a statistical (observational) construct, but a causal one [8,15]. It is of fundamental importance for definitions of causality that consider *X* to be a direct cause of *Y* if and only if the value (or, more generally, the conditional probability distribution) of *Y* differs for different values of *X*, holding all other potential causes of *Y* fixed [16].

### 4.2. Conditional Independence and Potential Causalityin BNs

Arrows in a BN are *causally oriented* if direct causes point into their direct effects, but this begs the question of what is meant by a direct cause (or its direct effect) and how they are to be identified from data. Algorithms for causal discovery propose constructive answers to these questions and clarify what causal questions can and cannot be answered from observational data [8,17]. The undirected arc between *Father* and *Mother* in Figure 4 signifies that they are informative about each other (e.g., because men and women of similar heights are more likely to marry), although perhaps neither can be interpreted as a cause of the other. More generally, arrow directions in non-causal probabilistic graphical models are often best regarded as arbitrary, since mutual information between random variables is symmetric: if *P*(*x*|*y*) differs from *P*(*x*) (conditional probability does not equal marginal probability), then *P*(*y*|*x*) = *P*(*x*|*y*)*P*(*y*)/*P*(*x*) = [*P*(*x*|*y*)/*P*(*x*)]**P*(*y*) must also differ from *P*(y). Just as Bayes’ Rule allows the joint probability *P*(*x*, *y*) to be decomposed equally well as either *P*(*x*|*y*)*P*(*y*), corresponding to the DAG model *Y* → *X*; or as *P*(*y x*)*P*(*x*), corresponding to the DAG model *X* → *Y*, so arrow directions in a fully specified Bayesian network (BN) only show one way to decompose a joint probability distribution into marginal distributions and CPTs. Other equally valid decompositions are usually possible. Thus, if suggested causal interpretations for arrows are provided, they must rest on principles other than conditional independence or mutual information, as these are symmetric.

Yet, there is a close relationship between conditional independence and most concepts of causality. Intuitively, a direct cause of a variable provides information about it that cannot be obtained from other direct causes. Thus, a variable is not conditionally independent of its direct causes, even though (by definition) it is conditionally independent of its more remote ancestors, given the values of its parents. These considerations often stop short of uniquely identifying the direct causes of each variable because conditional independence tests alone may not be able to distinguish between the parents and children of a node. For example, the three DAGs *X* → *Y* → *Z*, *X* ← *Y* → *Z*, and *Z* → *Y* → *X* have the same conditional independence relationships among variables, with *X* and *Z* being conditionally independent, given *Y*. They are indistinguishable by conditional independence tests (i.e., they belong to the same “Markov equivalence class”), but the constraints imposed by conditional independence do provide a useful filter: if no variable is conditionally independent of its direct causes, then statistical tests for conditional independence can help identify *potential* direct causes, even if they cannot always distinguish direct causes from direct effects. Conditional independence tests are built into current causal discovery and BN structure-learning software packages, such as *CompareCausalNetworks* [17] and *bnlearn* [18]. Although conceptual counterexamples can be contrived, e.g., by making *X* identical to *Y* in *X* ← *Y* → *Z*, so that Z is conditionally independent of *Y* given *X*, even though *Y* rather than *X* is a direct cause of *Z*, practical workers might be willing to disregard such special cases, perhaps regarding them as having a probability measure of 0 or close to it in real datasets, in order to get the benefits of automated screening for potential direct causes based on an (almost) necessary, although not sufficient, condition. For example, in the above Markov equivalence class, conditional independence implies that *X* is not a direct cause of *Z* (since *Y* separates them, i.e., conditioning on *Y* renders them conditionally independent) and that *X* and *Z* are not both causes of *Y* (since conditioning on *Y* renders them independent rather than dependent), although it leaves ambiguous whether *Y* is a cause or an effect of *Z*.

### 4.3. Causal Discovery for Predictive, Interventional, and Mechanistic Causation

Causal discovery algorithms apply additional principles beyond conditional independence to disambiguate possible directions of causality (if any) between variables. Useful principles for orienting mutual-information links between variables incorporated in modern causal discovery packages include the following [17].

**V-structures (explaining away):** If *X* and *Y* are unconditionally independent, but become dependent after conditioning on a third variable *Z*, then *Z* is a common child (or possibly a common descendant, if there are other variables) of *X* and *Y*. This is called a “collider” or “v structure” DAG: *X* → *Z* ← *Y*. For example, if *Mother* and *Father* heights are independent variables, but the adult offspring *Height* depends on both of them, then observing a tall person with a short father might make it more likely that the mother is tall, thus inducing a negative correlation between parent heights.

**Directed information flow and predictive causation (Granger) vs. manipulative causation:** Information flows from causes to their effects over time. Parents’ heights are potential causes of offspring heights, but not *vice versa.* This intuitively appealing idea leads to tests for *predictive causality*, such as classical Granger causality testing in time-series analysis and its nonparametric generalization to transfer entropy [19]. In these tests, *X* is defined as a predictive cause of *Y* if future values of *Y* are not conditionally independent of past and present values of *X*, given past and present values of *Y* itself. An intuitive motivation is that changes in causes help to predict changes in their effects [20]. However, predictive causation is a relatively weak causal concept: in a dataset that does not contain smoking as a variable, nicotine-stained fingers might be a predictive cause of lung cancer without being a *manipulative (*or “*interventional*”*) cause*, i.e., *X* being a predictive cause of *Y* does not necessarily imply that intervening to change the value (or distribution) of *X* would change the probability distribution of *Y*. Interventional causation, in turn, is weaker than *mechanistic causation*: in a world where fingers can only be kept clean by not smoking, nicotine-stained fingers could be a manipulative cause and a predictive cause of lung cancer without being a mechanistic cause. For causally effective decision-making, manipulative causation is necessary and sufficient, but it cannot be inferred from conditional independence and directed information flows alone.

**Dynamic Bayesian networks (DBNs)** make the flow of information between variables over time explicit by replacing individual variables with time-stamped values, creating a time series of values for each variable in successive periods. The conditional probability distribution for each variable in each period is allowed to depend on past values of other variables, possibly including its own lagged values.Causal discovery algorithms for learning DBNs from both experimental [21] and observational data [22] have been developed (e.g.,CaMML package, https://bayesian-intelligence.com/software/BI-CaMML-Quickstart-Guide-1.4/ (accessed on 19 April 2021).

**Effects are often simple (e.g., approximately linear) functions** of their direct causes andof random noise, as in the regression model in Equation (1). This provides a basis for statistical tests for the direction of causality between variables if noise is not Gaussian [23,24].

**Causal laws (CPTs) are invariant across applications contexts.** If a CPT *P*(*Y*|*Pa*(*Y*)) represents a causal law, then it should be the same across contexts, meaning that the same inputs (parent values) produce the same conditional probability distribution for the output *Y*. The property of *invariant causal prediction* (ICP) stipulates that the CPT is the same across studies (causal generalization) and across interventions (transportability), where an intervention sets values of some of the parents [25,26]. This property can sometimes be tested if the effects of interventions are available from multiple studies with different values of *Pa*(*Y*). For example, adding a study identifier code to other potential predictors of *Y* lets a CART tree or other conditional independence tests check whether the distribution of *Y* is conditionally independent of the study ID (and hence the specific interventions, if any, in different studies) once values of *Pa*(*Y*) are known [26].

These principles and algorithms have helped to identify plausible structures for causal BNs in many domains [22], but they often give somewhat different answers. Some require strong assumptions, such as that there are no unmeasured common causes of the observed variables (e.g., hidden confounders) that, if known, would explain away the observed dependencies between measured variables. Even whether one variable is a direct or an indirect cause of another in a dataset may depend on what other variables are measured. For example, the father’s height might be a direct cause of the offspring’s height in a dataset consisting of *Father*, *Mother*, *Gender*, and *Height*, and yet the two might be conditionally independent in a dataset that also contains more mechanistically relevant information, such as the offspring’s levels of human growth hormone during development until adulthood.

### 4.4. Knowledge-Based Constraints Help to Orient Arrows to Reflect Causal Interpretations

Current causal discovery algorithms are often unable to establish unique directions for arrows based on passive observations alone. For example, Figure 5 shows the results of applying four current structure-learning algorithms to the heights dataset (hill-climbing (hc), tabu search, grow–shrink (gs), and incremental association Markov blanket (iamb); see [18] for details of these algorithms). Only two of them (hc and tabu) oriented an arrow from *Father* to *Height*; the other two algorithms reversed this orientation. Knowledge-based constraints can help orient arrows to reflect causal interpretations. For example, if it is known that a variable is a sink in a causal BN, meaning that it can have parents (representing direct causes) but not children (e.g., *Height* might be an effect of *Father*, *Mother*, or *Gender*, but not a cause of any of them), then classification and regression tree (CART) algorithms [27] with the sink as a dependent variable provide useful heuristics for identifying direct causes and estimating CPT from the data. Figure 6 shows an example for *Height*. The CART algorithm (implemented in the *rpart* package in R) identifies *Height* as depending directly on (i.e., not being conditionally independent of) *Father*, *Mother*, and *Gender*. Their relative “importance” scores are estimated as 15, 6, and 80, respectively, based on their contributions to predicting *Height*; see the *rpart* package documentation and [27] for details of CART importance scores. Likewise, after imposing the knowledge-based constraint that *Height* is a sink, all four of the BN-learning algorithms used in Figure 5 correctly identify *Father*, *Mother*, and *Gende*r as parents of *Height*.

**Figure 5 entropy-23-00601-f005:**
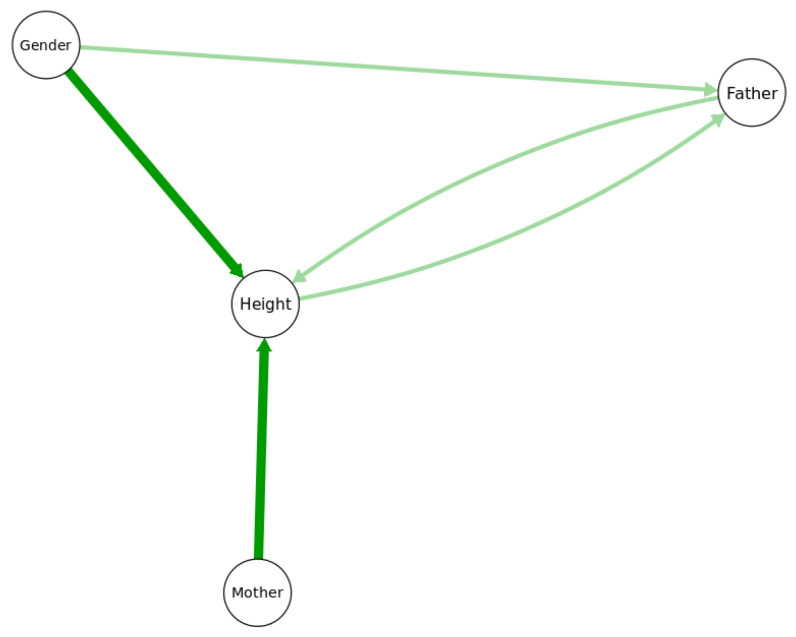
Bayesian network (BN) structures learned by four different algorithms in the *bnlearn* package only partly agree. All 4 algorithms agree on the heavy arrows (darker color), but only 2 of 4 agree on each thin arrow (lighter color). (Continuous variables were discretized using the Hartemink discretization pre-processing option, and hc, tabu, gs, and iamb algorithms were applied).

Figure 6 displays CPT information for *Height*, summarized as boxplots for the empirical conditional distributions of *Height* in the leaf nodes at the bottom of the tree. Each leaf node yields a different conditional probability distribution for *Height*, given the values or ranges of values shown on the branches for the “splits” (answers to binary questions) leading from the root node at the top of the tree to the leaf node. These splits define each leaf node and the information on which its probability distribution for *Height* is conditioned. For example, a male (M) with Father ≥ 68.35 inches and Mother ≥ 67.5 inches has a conditional distribution for *Height* summarized by the boxplot in the right-most leaf node, with a median value of over 72 inches. There are 22 such cases in the dataset (displayed as *n* = 22 for node 11). BN-learning algorithms, such as those in R packages *bnlearn* and *CompareCausalNetworks*, provide multiple methods for performing conditional independence tests and identifying the parents and estimating the CPTs of sink nodes, but the CART tree heuristic remains one of the simplest and most interpretable. If variables are not conditionally independent of their direct causes, then CART trees for sink nodes show their direct causes, provided that the differences they make in the conditional distribution of the sink variable are large enough to be detected by the tree-growing algorithm.

**Figure 6 entropy-23-00601-f006:**
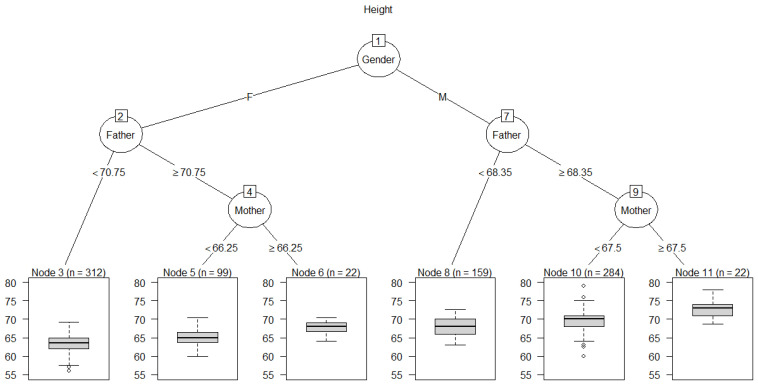
A Classification and regression tree (CART) for *Height* showing boxplots for the conditional distributions of *Height* given *Gender*, *Father*, and *Mother*. Dots in the boxes represent outliers.


# R code for Figure 6
library(rpart); library(rpart.plot); library(partykit); tree <- rpart(Height~Father + Mother + Gender);
plot(as.party(tree), type = "extended", main = "Height")

### 4.5. Structure of Most Probable Explanations (MPEs) in Bayesian Networks

In a probabilistic DAG model with causally oriented arrows, a *direct explanation* for an observed value states that *Y* has the value it does because the parents of *Y*, denoted by *Pa*(*Y*), have the values they do. These values entered the CPT for *Y*, which can be denoted as *P*(*Y*|*Pa*(*Y*)) (or more explicitly as *P*(*Y* = *y*|*Pa*(*Y*) = *x*), where *y* denotes a specific value of *Y* and *x* a specific vector of values for *Pa*(*Y*)) to generate the observed outcome for *Y*. Alternatively, *indirect explanations* for the value of *Y* state that the parents of *Y* have the values they do because of their own parents and CPTs. One can, therefore, at least partially explain an *a priori* improbable value of *Y* by finding values (or combinations of values) of its parents that, when conditioned upon, make the observed value of *Y* more probable. Such explanations in causal DAG models can be deepened recursively: values of variables are explained by explaining the values of their parents. This extends the concept of explanation beyond the statistical regression concept of proportion of variance of a dependent variable explained by differences in its explanatory variables. A causal DAG model makes probabilistic dependencies among the explanatory variables explicit, leading to a hierarchy of explanations rooted in the values of input variables (nodes with only outward-pointing arrows) and in chance at the CPTs. Explanations in such models are not limited to explaining factual observations. A standard application of Bayesian inference algorithms in BNs is to find values for unobserved variables to maximize functions of the observed data, such as the (log) likelihood function, the posterior probability, or a generalized Bayes factor of the observed values, thereby generating *most probable explanations* (MPEs), *maximum a posteriori* (MAP) explanations, and *most relevant explanations* (MREs) for the observed data, respectively [28]. These algorithms are also frequently applied to hypothetical, counterfactual, and potential future events, rather than to observed ones. For example, if it is posited that a complex system eventually fails, then these BN “explanations” (i.e., settings of the values of unobserved variables to maximize an objective function), given the hypothesized failure information, can be used to help envision how the failure might occur, which may help to prevent it. A combinatorically complex problem is to find the simplest such explanations (by various criteria, e.g., with irrelevant conditions pruned away). Exact and approximate methods have been developed for this purpose [29]. However, all these methods are restricted to passive explanations: they explain some (real or imagined) observations in terms of other observations, rather than in terms of design decisions or other active decisions and interventions.

### 4.6. Explaining Predictions for Effects of Interventions: Adjustment Sets, Causal Partial Dependence Plots (PDPs) and Accumulated Local Effects (ALE) Plots for Known Causal BNs

In a causal BN, exogenously changing the value of one variable changes the conditional probability distributions of variables into which it points (its direct effects) via their CPTs. Changing their distributions, in turn, changes the distributions of their children. The effects of the initial change propagate along causally oriented arrows, leading to updated aleatory distributions for its descendants. The process is analogous to the propagation of evidence in a non-causal BN but describes propagation of *changes* in aleatory probabilities in response to actions, rather than (or in addition to) propagation of *inferences* about epistemic probabilities in response to observations [8]. In this setting, the statistical challenge of estimating how changing one variable would change the distributions of others can be met using a combination of graph-theoretic and statistical estimation methods. Specifically, to predict how changing *X* (via an exogenous intervention or manipulation) would change the distribution of one of its descendants, *Y*, it is necessary to identify an *adjustment set* of other variables to condition upon [30]. Adjustment sets generalize the principle that one must condition on any common parents of *X* and *Y* to eliminate confounding biases, but must not condition on any common children to avoid introducing selection biases [19]. Appropriate adjustment sets can be computed from the DAG structure of a causal BN for both direct causal effects and total causal effects [30]. Then, given a dataset of observed values for all variables for each of many cases, the quantitative effect of changes in *X* on the conditional mean (or, more informatively, the conditional distribution) of Y can be displayed via a *partial dependence plot* (PDP) [13]. This shows how the conditional distribution for *Y* (or its mean and uncertainty bands), as predicted by a machine-learning technique (e.g., random forest, support vector machines, gradient boosted machines, deep learning networks, etc.), changes for different values of *X*, holding the values of variables in the adjustment set fixed at the values they have in the cases in the dataset. We call the result a *causal PDP*. It shows how *Y* changes with *X* when sources of bias (e.g., possible confounders) are controlled by holding their values fixed, but other variables are allowed to adjust realistically as *X* varies. For example, in a causal BN model *X* → *Z* → *Y*, the probability distribution of *Z* would change as described by its CPT in response to changes in the value of *X*. The distribution of *Y* would then change in response to the change in the distribution of its input, *Z*. If all relevant confounders are measured and controlled in this way, then the changes in *Y*’s distribution are presumably caused by the differences in *X* values [16,19].

In this causal BN framework, the predicted change in *Y* in response to a change in *X* is explained in terms of the following components.

**a.** **Probabilistic causal laws**, represented by CPTs. These causal CPTs remain fixed (“invariant”), independent of the situation or interventions to which they are applied [25] and of the particular studies from which they were derived [2]. In this sense, a CPT in a causal BN encodes a *causal generalization* from specific studies and experiments. It describes how aleatory probabilities for a variable’s values depend on the values of its parents. The invariance of this description across settings and interventions, termed *invariant causal prediction* (ICP), reflects the universality of causal laws (*ibid*). (In a non-causal BN, by contrast, CPTs are statistical description of conditional probabilities that may change across applications, perhaps reflecting different mixtures of aleatory CPTs based on different values of causally relevant latent variables.)**b.** **Initial conditions** to which these laws are applied. These conditions may differ across applications. For example, an exogenous intervention that sets *X* to a new value changes the conditions entering the CPTs of its children, initiating a change in their conditional distributions that then propagates along causally directed paths to change downstream distributions. The same intervention can have different effects in different populations if other conditions are also different between them. Thus, an infection control intervention that greatly reduces infection-related mortality rates in one hospital might not do so in another with a different mix of patient conditions, not because the same causal laws (aleatory CPTs) don’t hold, but because the distributions of other variables that affect success (i.e., other causal parents of success) differ. In such cases, *transportability formulas* show how to adjust the findings from one set of conditions to apply to a new set of conditions, if there is enough common information between them so that this can be done [31]. For example, if the success of the infection control program varies by the age and sex of patients (and perhaps other covariates, which would be parents of the success indicator variable in a BN model), then the conditional success probabilities for different combinations of age and sex and any other causally relevant covariates can be applied to the empirical joint distributions of these variables at other hospitals to predict how well the program will succeed elsewhere and explain why.**c.** **Causal network structure**, i.e., the topology of the causal BN, showing the direct causes that each variable depends on, i.e., its parents in the causal BN, if any.**d.** **An adjustment set** specifying which variables are held fixed in calculating the causal effect of a change in *X* on changing the distribution of *Y*. There are often multiple valid adjustment sets and comparing the PDPs for the estimated causal effects of changes in *X* on *Y* across alternative adjustment sets provides an internal validity consistency check for whether the assumed BN model provides a consistent description of the causal effects [30].**e.** **Assumed values** for the variables that are held fixed. In PDPs, the assumed values are those that the variables have in the dataset, used to estimate the PDP.

A causal PDP curve plotting the conditional expected value of *Y* for each value of *X*, with uncertainty bands included if desired, provides a concise summary of how *Y* is predicted to change if *X* is changed, holding variables in the adjustment set fixed at specified levels. Rather than answering “*If we observe that Y = y, what is the most probable explanation?*” as for MPEs, a causal PDP answers “*If we set X = x, how will the expected value (or, more generally, the conditional distribution) of Y change?*” given stated levels of other variables. Fully explaining the answer requires the preceding five components, (a)–(e). The resulting ability to make (and to explain) predictions about effects of interventions provides essential information for optimizing decisions.

The graph-theoretic part of the causal BN explanatory framework provides rigorous conditions and algorithms for identifying adjustment sets and transportability conditions and formulas, when they exist [30,31]. The quantitative estimation and display of average causal effects by PDPs is less satisfactory because averaging effects over different individual cases and hypothetical conditions obscures exactly what is (and should be) assumed about each case [32,33].The concept of “holding fixed” some variables to isolate the causal effects of others becomes problematic when the variables are tightly coupled, so that changing one entails changing others, rather than holding them fixed. For example, suppose that heart attack risk depends on both height and weight, and that weight also depends on height. To isolate the effect of weight on heart attack risk, a causal PDP would first include height in an adjustment set, and then modify the value of weight in each record of a dataset (i.e., each case), setting it equal to different values while holding heights fixed at the values they have in the records. The PDP would record the average predicted heart attack risk for each value of weight. The needed predictions are made by applying a predictive ML method, such as random forest, to each record for each value of weight, holding other variables fixed. In reality, weight and height are not independent. Averaging predicted risks over hypothetical cases with modified values of weight that are unlikely or impossible in light of the corresponding heights undermines the practical relevance of the resulting PDP by diluting it with predictions for unrealistic hypothetical cases that fall outside the observed range of observed data values. To avoid this limitation, *accumulated local effects* (ALE) plots do not involve counterfactual conditions outside the range of the data, but instead quantify how small (“local”) changes in one variable would affect the predicted value for another for each individual (again using a predictive model such as random forest) [33]. The effect of a small change in *X* on the expected value of *Y*, evaluated at *X* = *x*, is averaged over cases with *X* approximately equal to *x*, rather than being averaged over all cases (by setting *X* = *x* for all cases, while holding other variables fixed, as in PDPs). Thus, instead of asking how heart attack risk would differ for a 7 foot man if his weight were 100 pounds instead of 300 pounds, the ALE asks how heart attack risks would change for each individual if his or her weight were reduced by 1 pound. It displays the answer for individuals with different weights. Similarly, *individual conditional expectation* (ICE) plots [32] can be used to show how *E*(*Y*|*x*) varies with *x* for each individual case, thus avoiding the ambiguities of aggregation. These ICE plots, in turn, can be clustered to reduce visual clutter.

Figure 7 illustrates these ideas, showing a PDP (upper left), ALE (upper right, y axis showing deviations from average), and ICE cluster plot (lower, y axis showing deviations from average) for the Heights dataset, using random forest as the prediction engine. (The plots were generated by the Causal Analytics Toolkit at http://cloudcat.cox-associates.com:8899/ (accessed on 19 April 2021), which automatically optimizes the number of clusters in the ICE cluster plot.) We refer the interested reader to [32] for ICE plots and to [33] for ALE plots. Both also discuss PDPs. Apley and Zhu note that both PDPs and ALEs correctly recover the additive components in generalized additive models. To our knowledge, no software package currently disaggregates ALE curves to the individual level, analogous to ICE plots for PDPs, but this seems to be a natural next step.

**Figure 7 entropy-23-00601-f007:**
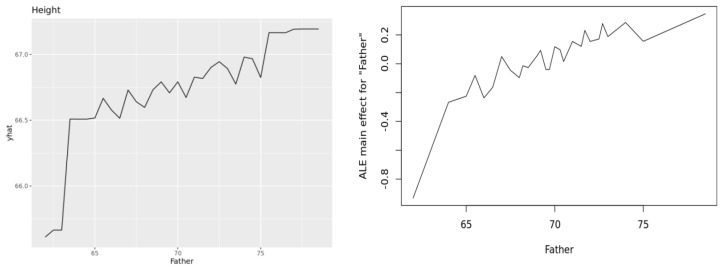
PDP (upper left), ALE (upper right), and ICE cluster plots (lower plot) for effect of Father height (*x* axes) on adult offspring heights (*y* axes). Plots are produced using the *pdp*, *ALEPlot*, and*ICEbox* packages in R, respectively, by the Causal Analytics Toolkit at http://cloudcat.cox-associates.com:8899/ (accessed on 19 April 2021).

These various approaches to defining, quantifying, and visualizing effects of one variable on another in a multivariate context are complicated mainly by conceptual ambiguities about what “effect” should mean when there are interactions among variables and heterogeneity in their values across cases. For direct causal effects in individual cases, there is no such ambiguity in the causal BN framework: the causal CPT fully describes how each variable’s aleatory probability distribution depends on, and varies with, the values of its parents. The CPT *describes* probabilistic dependency, rather than offering a deeper (e.g., mechanistic) explanation of why it exists and how it functions, but this description suffices to predict effects of interventions. For this purpose, a simple descriptive causal BN model *X* → *Y* for predicting or controlling *Y* via observations and interventions on *X* is as useful as a more refined causal BN model *X* → *Z* → *Y*, describing how *Z* mediates the causal relationship between *X* and *Y* (via *P*(*y*|*x*) *=* Σ*_z_P*(*z*|*x*)*P*(*y*|*z*)). The model *Father* → *Height* is just as useful and valid as the more refined model *Father* → *HGH* → *Height*, indicating that human growth hormone (*HGH*) mediates the probabilistic dependence of *Height* on *Father*, for the purposes of predicting the effects of *Father* on *Height*. More generally, the most useful level of detail to include in a causal explanation for supporting decisions describes how feasible changes in controllable decision variables change probabilities of outcomes. Causal BNs and their extensions to include decision variables and evaluation nodes (influence diagrams, discussed next) support this level of detail, while leaving open scientific questions of how underlying causal mechanisms mediate the dependencies among variables that are described by causal CPTs.

## 5. Structure of Explanations for Decision and Policy Recommendations in Influence Diagrams (IDs): Maximizing Expected Utility with a Known Causal Model

Causal BNs can be extended to produce AI decision-support advisory systems that recommend *decisions* (also called choices, actions, interventions, or manipulations) and *policies* (also called *decision rules* or *strategies*, defined as mappings from available information to actions, or, more generally, to probability distributions over actions). Actions and policies are viewed as *controllable causes*, i.e., decision variables with values selected from a set of possibilities (the choice set) by the decision-maker. The CPT for a random variable includes the values of any decision nodes on which it depends, i.e., with arrows that point into it in a causal DAG model. Decisions do not fit smoothly into the formalism of probability theory, as they are not events (subsets of a sample space), but they can readily be incorporated into BNs by introducing *decision nodes* and a sink node called the *value* or *utility node* that evaluates the outcomes at other nodes. The *utility node* represents the von Neumann–Morgenstern utility function for evaluating outcomes. The variables that point into it are those that matter to the decision-maker in evaluating the consequences of decisions. The resulting augmented BN is called an *influence diagram* (ID) [7,34]. In an ID, any of several alternative actions can be taken at a *decision node* (or “*choice node*”), based on information about the values of nodes that point into it. It is often convenient to allow deterministic functions for some variables in IDs as well, e.g., body mass index is a deterministic function of height and weight. These may be regarded as special cases of CPTs, giving probability 1 to a specific output value for each combination of values of its parents. Policies or decision rules, mapping information to (possibly randomized) choices of actions, are implemented at choice nodes.

In the ID framework, a decision rule sets the value of a choice node. The distributions of variables that depend on the choice node are then updated based on their CPTs, and these updates are propagated through the ID, ultimately leading to a conditional distribution and an expected utility (EU) value at the utility node. Decisions are optimized by algorithms that select decision rules to maximize expected utility. For tabular IDs, with a few possibilities at each node and explicit tables showing how conditional probability distributions of random variables depend on their parents, the same Bayesian inference algorithms used to calculate posterior distributions of chance nodes given any set of observations can be repurposed to infer decisions that maximize expected utility [35]; see also [36] for related work). More computationally efficient algorithms have also been developed for solving for optimal decisions in IDs [34,35]. In an ID, the explanation for a recommended (optimized) decision rule is that it *maximizes expected utility*, given the information available when decisions are made. If the ID is known to be a trustworthy model of reality, then its decision recommendations are justified by the normative axioms of expected utility theory. If the ID is estimated from data and has uncertain validity, then maximizing estimated EU may still be a valuable heuristic, but other considerations involving model uncertainty may also become important.

Figure 8 presents an example based on a dataset showing levels of sales for customers receiving different levels of Facebook, YouTube, and newspaper advertising. The left side of Figure 8 shows the results of four BN structure-learning algorithms; all agree that *sales* depends on *YouTube* and *Facebook*, but not on *newspaper* advertising levels. The right side shows a nonparametric response surface for *sales* as a function of *YouTube* and *Facebook*. Its shape reveals that ads in each channel increase expected sales most when the other channel also has relatively many ads. That is, there is cross-channel positive interaction (synergy). Given these data and information on a marketing budget and advertising prices for each channel, what decision recommendation should be made for how to allocate the budget to maximize resulting sales, and how should this recommendation be explained and justified?

**Figure 8 entropy-23-00601-f008:**
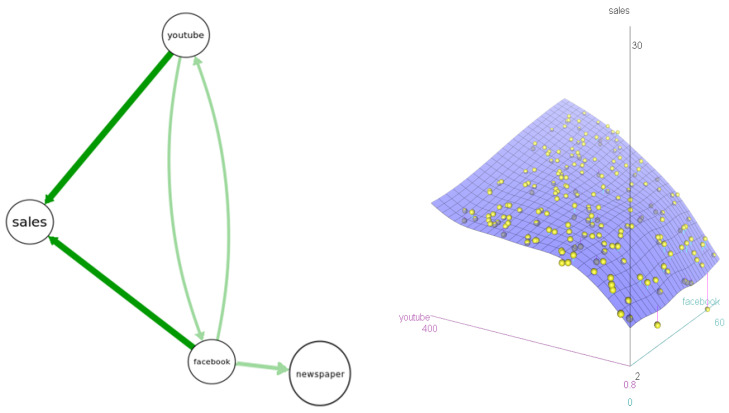
Left: Bayesian network (BN) structures learned by 4 algorithms in the *bnlearn*R package (hc, gs, tabu, iamb). Arrow thicknesses show how many algorithms (4 for darker-color arrows, 2 for lighter-color arrows) produce each arrow. Right: A response surface model for *sales* given *YouTube* and *Facebook*.


# R code for response surface
library(car); data("marketing", package = "datarium"); scatter3d(sales ~ youtube + facebook, fit = c("smooth"), data = marketing)

Figure 9 shows a standard managerial economics solution to this decision optimization problem. The solution is not related to standard ID solution algorithms for discrete decision variables, but instead applies nonlinear programming to choose the point satisfying the budget constraint that achieves the highest possible sales (the point, indicated by a star, where the budget line and an iso-sales contour are tangent). For simplicity, we assume that the goal is to maximize sales, or that the budget constraint is binding. The budget line is the set of feasible choices (the choice set for the decision problem) of all affordable combinations of *Facebook* and *YouTube* advertising that spend the budget, given the prices of advertising in each channel (which are divided into the budget to determine the intercepts of the line). Contours for equal predicted sales (the “iso-sales” contours, in microeconomics parlance) are estimated from the data by applying PDP estimation algorithms to the direct causes of sales, averaging over the fixed levels of other variables.

**Figure 9 entropy-23-00601-f009:**
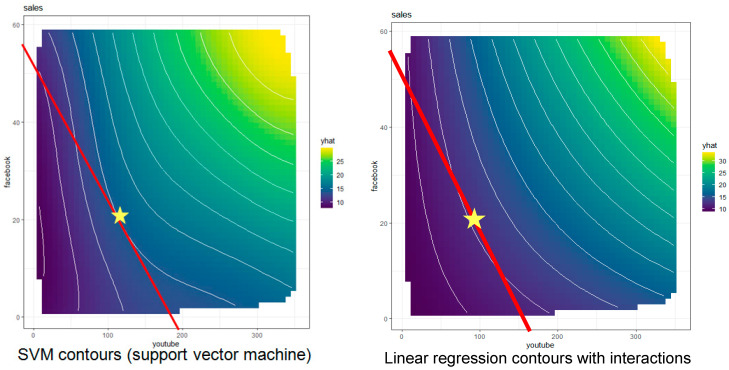
Estimated causal partial dependence plot (PDP) contours for sales based on support vector machine (**left**) and quadratic regression (**right**) with superimposed budget lines and estimated optimal budget allocations (stars) for each estimated causal PDP.


# R code for partial dependence plots
data("marketing", package = "datarium"); library(e1071); library(pdp); library(ggfortify)
svm<- svm(sales ~ ., data = marketing); lm<- lm(sales ~ .^2, data = marketing)
pdp1 <- partial(svm, pred.var = c("youtube", "facebook"), chull = TRUE); autoplot(pdp1, main = "sales", contour = TRUE); pdp2 <- partial(lm, pred.var = c("youtube", "facebook"), chull = TRUE); autoplot(pdp2, main = "sales", contour = TRUE) 

The left panel of
Figure 9
shows the contours estimated by a support vector machine (SVM) algorithm, and the right panel shows the corresponding contours estimated by multiple linear regression with second-order interactions. (In the right panel, the location of the sales-maximizing allocation is interpolated from the contours above and below the budget line, since none of the displayed contours is tangent to it.) Numerous other machine-learning techniques could be used equally well to estimate these contours, e.g., random forest, multiple adaptive regression splines (MARS), gradient boosted machines (GBM), and so forth. They give somewhat different estimates of the contours, and hence somewhat different recommendations for the optimal budget allocation. These differences reflect *model uncertainty:* the true contours are unknown, and the estimated contours depend somewhat on the specific estimation technique used. However, the causal structure for decision recommendations is clear: select the feasible decision (or combination of decisions, if we view expenditures in each channel as distinct decision variables) to maximize the expected utility of consequences (here, the expected resulting sales). Moreover, to the extent that the relevant causal model can be estimated from data, it is also clear how the optimal decision recommendation should change as conditions change. If budget or advertising prices change (altering the slope and intercepts of the budget line), or if the contours of the PDP change (e.g., because of events that affect the advertising effectiveness of different channels, perhaps by changing the reputations, credibility, or market penetrations of the parent companies), then the new point of tangency between the budget constraint line and the iso-sales contours identifies the new recommended decision for the altered circumstances.

In this way, the optimization-based framework can cope with limited novelty, offering defensible decision recommendations for what to do under new pricing, budget or effectiveness conditions, even if no historical data are available for the new conditions. The underlying causal model (the PDP contours, corresponding to a causal CPT for *sales,* given values of its parent decision variables, *YouTube* and *Facebook*, and averaging over the values of all other variables that might affect individual purchase decisions) provides the knowledge needed to justify decision recommendations even under new conditions. In this way, causal AI avoids the need for new training data to make and explain decision recommendations if causal CPTs satisfying ICP are available for predicting and optimizing effects of interventions under current, possibly novel, conditions. However, *model uncertainty* about the relevant causal CPTs, illustrated by the different estimates of contours in
Figure 9, leads to uncertainty about the best decision recommendation. It creates a need to learn more about the shape of the response surface (or causal CPT) over time to enable higher-value decisions. Reinforcement learning (RL) algorithms, discussed below, address this need.

## 6. Structure of Explanations for Decision Recommendations Based on Monte Carlo Tree Search (MCTS) and Causal Simulation

Making and explaining current decisions in terms of probabilities and utilities for their potential consequences epitomizes rational, deliberative, planning and decision-making for reason-driven “System 2” decisions, as contrasted with quick, pattern-driven “System 1” decisions [37,38]. When a known probabilistic causal model capable of predicting outcome probabilities for different actions is available, along with a utility function for evaluating different outcomes, then solving for expected utility-maximizing policies, even in multi-period settings, is conceptually similar to solving the simple nonlinear programming optimization problem illustrated in Figure 9. The relevant optimization techniques for stochastic dynamic systems are more complicated, e.g., with stochastic dynamic programming instead of simple nonlinear programming, but the principle is similar in that current decisions are optimized to maximize expected utility, taking into account future decisions and outcome probabilities [39]. For example, a Markov decision process (MDP) is a multistate transition reward process, for which a policy specifies which feasible action to take in each state. Actions can affect rewards from transitions as well as next-state transition probabilities. MDPs with finite numbers of states and actions can be solved for optimal policies that maximize expected discounted rewards or average rewards per unit time by using either linear programming, dynamic programming, or well-known special-purpose iterative algorithms [9,10]. As in Figure 9, re-optimizing when conditions change allows new optimal policies and resulting decision recommendations to be created as needed for new situations; they are justified by optimization.

The optimization-based paradigm for recommending and justifying decisions must be extended if the objective function to be optimized is unknown, either because it is difficult to compute (perhaps requiring lengthy simulation) or because of a lack of causal information about how actions affect outcome probabilities. If uncertainty about an MDP is described by convex uncertainty sets for transition matrices—e.g., if the one-step transition probabilities are known to lie in certain intervals, or if the rows of the transition matrix are known only to lie within some relative entropy (Kullback–Leibler) “distance” from known reference distributions—then MDP solution algorithms can be extended to find the best policy for the worst-case model in the uncertainty set, providing a form of robust control for uncertain MDPs [40]. (Extensions of expected utility theory that imply such maximin decision-making over convex uncertainty sets in decision analysis are discussed by [41]) However, the optimization paradigm breaks down if the optimization problem is unsolvable, e.g., because optimal policies are incomputable. This happens in some partially observable MDPs (POMDPs), i.e., if the states of MDPs cannot be directly observed (so that policies mapping states to actions cannot be implemented) but must be inferred from observations that depend probabilistically on the underlying state—e.g., from signals observed via noisy information channels in engineering systems; from observed symptoms in patients with underlying unmeasured health conditions; from observed behaviors of customers with underlying unmeasured beliefs, preferences, and intents; and so forth. Existence of optimal policies—or even of policies meeting desired constraints, such as having at least a stated probability of achieving a goal state in finite time without first encountering any catastrophic state—is undecidable for some POMDPs with infinite planning horizons, and with either discounted or undiscounted total reward models [42]. Although important special cases are decidable [43], e.g., by converting the POMDPs to equivalent dynamic Bayesian networks (DBNs) and then optimizing policies by maximizing the likelihood of suitably normalized rewards [44], finding optimal or near-optimal policies in many POMDPs, even when it is possible in principle, is sufficiently computationally complex so that heuristics must be used [43].

The rationale for heuristic-guided decision or policy recommendations in such cases is that *no better choice has been discovered by the time decisions must be made.* This “any time planning” principle applies more generally, well beyond the class of POMDP decision problems. For example, several recent state-of-the-art probabilistic planning heuristics apply sampling-based methods, such as Monte Carlo Tree Search (MCTS) with deep learning or other function approximators of value functions, to search for and incrementally improve the best (highest expected value) plan that can be discovered with a given computational budget. When an action must be taken, the best plan discovered so far is used to recommend what to do next [45,46,47]. Other causal simulation and simulation-optimization algorithms [48,49], e.g., particle filtering for efficient sampling of decision and outcome trajectories in POMDPs with hidden (latent) variables, differ in detail from MCTS, but they use similar principles of selective sampling and expansion (via simulation) of future trajectories for decisions and outcomes to estimate value functions and heuristically optimize immediate decisions [50,51]. We will refer to such methods generically as *causal simulation heuristics*, as they use causal models to envision a sample of future scenarios that are then used to evaluate and choose among currently feasible courses of action. Causal models that describe probabilistic relationships between actions and consequences (next states and rewards) are crucial inputs to these methods. They are used under the guidance of MCTS, particle filtering, or other (e.g., hybrid MCTS-particle filtering) causal simulation heuristics to efficiently sample, simulate (“roll out”), and evaluate possible futures. The resulting information is used to improve plans specifying what to do now and what to do next if various contingencies occur [50]. Re-planning and searching for better plans occur as new information and additional time for planning are gained, with decisions at any time being guided by the current best plan. Causal simulation guided by entropy-based upper bounds on the value of the best policy (maximum entropy MCTS) converges quickly (exponentially fast) to the optimal next decision for some classes of sequential decision problems for MCTS [47].

## 7. Structure of Explanations for Decision Recommendations Based on Reinforcement Learning (RL) with Initially Unknown or Uncertain Causal Models

If a causal model specifying how actions affect outcome probabilities (e.g., reward and state transition probabilities in an MDP) is not available, a key new principle is needed to guide decision-making. When causal models and optimal policies are initially uncertain, *actions are valued not only for the rewards and state transitions that they cause, but also for the value of the information that they reveal* about how to improve policies [52]. Managing the famous exploration–exploitation tradeoff between applying the most promising policy discovered so far (exploitation), and deviating from it to discover whether a different policy might perform better (exploration) requires taking into account the value of information (VoI) produced by actions (*ibid*). For example, in a multiarm bandit (MAB) problem, such as a clinical trial of several alternative treatments where the efficacy of each treatment is initially unknown, a simple Bayesian heuristic starts with the uniform prior probability that each treatment is best. It updates these probabilities by conditioning on observed results [53]. The convenient fact that the family of beta distributions, which includes the uniform, is a self-conjugate family for binomial sampling yields simple update formulas (posterior beta distributions) for MABs with fixed but initially unknown probabilities of success for each treatment. More complex update formulas and approximations have been developed for MABs with more complex reward distributions [54]. A simple but very effective technique for balancing exploration against exploitation in this setting is to probabilistically select, on each trial, each treatment with a probability equal to its current probability of being best. Thus, a treatment known to be best would always be selected; one known not to be best is never selected; and treatments deemed more likely to be best are selected more frequently (“reinforced”) as experience accumulates. This simple *reinforcement learning* (RL) heuristic [55], known as Thompson sampling (TS), has been extended to MDPs with the number of periods (the “decision epoch” length) for which each trial policy is evaluated being tuned to be long enough to evaluate the policy without a needlessly inefficient search [53]. Variations of Thompson sampling are asymptotically optimal (converging in the mean to optimal-value policies) even for many non-Markov and partially observable environments [56].

Performance guarantees for TS and other RL algorithms are usually given in terms of asymptotic optimality and sublinear growth of regret. The *regret* from using a policy *P* over any time interval is defined as the expected difference between the sum of rewards that would be obtained by acting optimally throughout the interval, as assessed by someone with perfect information about how actions affect outcome probabilities, and the sum of rewards actually obtained by following policy *P*. If regret grows sublinearly with the length of the interval, then the policy being used must be optimal (zero regret) or getting closer to it as time passes, since otherwise regret would increase at least in proportion to the length of the interval over which it is accumulated. Sublinear bounds and asymptotic zero-regret guarantees for the performance of RL algorithms, including TS, have been established for MDPs, MABs, and other classes of sequential and adaptive decision problems [53,57]. Recent refinements include sampling each action to minimize the ratio of the square of expected error (i.e., of squared one-period regret) per unit of information gained about how to optimize choices (i.e., per bit of mutual information between the optimal action and the next observation). This heuristic outperforms Thompson Sampling and other heuristics across a wide class of models, with a bound on regret that scales with the uncertainty (entropy) of the optimal action distribution [58]. If the pace of change is slow enough, then RL algorithms yield low-regret policies even for MDPs with drifting (i.e., slowly changing, in a sense that can be made precise) parameter values [59]. If risk is important, imposing safety constraints on learning to avoid sampling actions with potentially catastrophic consequences still allows zero-regret policies to be learned in some MAB problems [60,61]. Risk-averse dynamic programming methods that protect against large negative deviations from expected values have been developed for MDPs using average value-at-risk (VaR) [62,63]. Other coherent risk measures, including conditional-value-at-risk (CVaR) and entropic-value-at-risk (EVaR), have recently been used in risk-averse planning and decision optimization algorithms for both MDPs and POMDPs [64,65], as well as in model-free RL algorthms [66].

Other refinements, extensions, and improvements in RL algorithms for various classes of decision problems continue to be made, as this is a very active area of ongoing research [55,57]. For example, in nonlinear optimization problems, such as optimizing cross-channel advertising expenditures in Figure 9, the *smoothness* of the estimated response surface, as exhibited in its contours, allows valuable information about the best policy (feasible allocation of budget) to be gleaned from observations at nearby points. Collecting information in the vicinity of the current estimated optimum to better ascertain the shape of the response surface, thereby enabling incremental adjustments of the decision variables to increase mean system performance, was one of the earliest approaches to decision-making with uncertain response surface models [67]. MABs have been extended to allow for such spatial correlations among rewards at different locations in a decision space, and RL algorithms have been developed for the safe exploration and optimization of policies with correlated rewards when some combinations of decision variables have low rewards (or high penalties) and must be avoided [1,68]. RL methods have also been integrated into MCTS heuristics for sequential planning and decision-making in MDPs, POMDPs, and more general probabilistic environments to guide efficient search for high-performance plans satisfying safety constraints [60,69]. They have been applied to POMDPs, to semi-Markov decision processes (SMDPs) and to partially observable SMDPs, in which executing an action (or the subtasks of which it is composed) takes an uncertain amount of time [70,71]. For this often realistic setting of tasks with uncertain durations and success probabilities, RL has also been integrated with behavior trees [72] and with hierarchical POMDPs [44,71] to enable AIs to adaptively select behaviors (sequences of tasks and subtasks) over time. Importantly, RL can be used to acquire and improve *skills*, i.e., learned sequences of actions for accomplishing tasks and subtasks [3,73]. Multiple skills and behaviors can be executed simultaneously as when an autonomous robot executes, in parallel, routines for moving toward a target while maintaining balance and avoiding collisions. This concept of choosing behaviors from a set that can be expanded by acquiring and perfecting new skills extends the usual decision–analytic model of decision-making, in which a decision-maker optimizes over a fixed choice set of possible alternative choices, by allowing investment in skills that expand the repertoire of behaviors that can be successfully executed in response to a situation. What it knows how to do (i.e., the current choice set) then forms part of the explanation for what an agent (whether an AI, a person, or a team or organization) chooses to do at any time [3].

Throughout these variously extended and generalized classes of decision problems, however, the basic RL principle of adaptively selecting actions and policies to reduce expected regret, considering the value of information as well as the value of rewards, has proved extremely valuable, not only for MABs, MDPs and POMDPs with initially unknown parameter values, but also for an increasingly wide variety of other dynamic stochastic systems with initial uncertainty about how actions affect transitions and rewards [56]. Simply adding an “entropy regularization” term, reflecting current uncertainty (entropy) about the best policy for an MDP, to the standard objective function (e.g., a weighted sum of rewards) in RL to encourage more random exploration when the best policy is more uncertain has been found to improve the robustness and convergence of RL for MDPs with continuous state and action spaces [74]. In effect, successful RL heuristics use adaptive exploration, guided by current uncertainty about the best (reward-maximizing) policy, to learn enough about the causal relationships between actions and consequences to enable low-regret policies to be discovered [75]. RL heuristics thereby provide a basis for decision recommendations along the way that can be explained and justified as contributing to a balanced mix of exploration and exploitation while meeting safety and risk constraints if necessary [60]. Although they are guaranteed to eventually learn optimal policies with probability approaching 1 in certain stationary environments, such as ergodic MDPs or MABs, RL algorithms may perform poorly in non-stationary environments, such as for “restless bandits” with reward distributions that change quickly compared to the convergence rate for learning low-regret policies [76]. More generally, in dynamic decision environments, the performance of an AI decision and control system is limited by the rates at which it can interact with and learn from its environment, as discussed next.

## 8. Limitations and Failures of Causally Explainable Decisions for Dynamic Systems

An AI system can fail to provide causally effective decision recommendations to achieve desired states and safety goals if it is overwhelmed by its environment—for example, if the conditional state transition or reward probability distributions in response to its actions change too quickly for the AI learning and decision algorithms to keep up, or if the AI’s repertoire of available actions is too limited to greatly change the course of events. Suppose that the state of a controlled system changes probabilistically not only in response to choices made by an AI controller, but also in response to stochastic inputs from the environment. The state of the system at any time is only partially observable by the AI controller via a noisy (probabilistic) information channel representing its limited sensing capabilities. In contrast to mathematical optimization formulations that treat choosing an action as a primitive that can be executed instantaneously, correctly, and with certainty, suppose that the AI controller can only select control signals to send to imperfect actuators (e.g., people or agents or mechanisms acting on its behalf). Implementation of its control signals as actions may involve errors and uncertainties. An intended action may be interrupted or fail before it is completed. Thus, the control channel is also probabilistic: a control signal sent by the AI creates a probability distribution over implemented actions (controllable inputs) entering the system and possibly affecting its future state. In this context, the AI’s ability to steer the system toward desired (highly valued) goal states while avoiding undesired ones is limited by the capacities of its sensing and control channels, i.e., the rate at which information about the state of the system can be received [77] for feedback control stabilization of stochastic linear systems) and the rate at which information can be transferred from control signals to future states of the system, called the *empowerment* or *control capacity* of the controller [78,79,80]. Thus, a scalar stochastic linear system with actuator uncertainty can be stabilized around a desired state if and only if its control capacity—which is used to dissipate uncertainty about the state of the system—is greater than the rate at which uncertainty about the state accumulates in the absence of control [80]. If the control capacity is below this limit, then even an arbitrarily “intelligent” AI, meaning one capable of instantly computing fully optimal policies with respect to the information it has and the control signals it can send, will lack power to implement causally effective policies to bring the system to (or close to) a desired state and maintain it there. Conversely, even if control and sensor capacities are high enough not to limit the performance of a controller, the difficulty or impossibility (undecidability) of computing optimal or approximately optimal polices for some problems, including some POMDPs with infinite horizons, limits the quality of decision recommendations that can be given for these problems [42,81]. In short, the value of decision recommendations that any system can give depends greatly on the decision problem and on the available causal model (if any) linking actions to outcome probabilities. Such models range from relatively simple tabular influence diagrams, response surface, and optimization models (Figure 8 and Figure 9), for which optimal recommendations are readily computed and defended, to undecidable infinite-horizon POMDPs or computationally hard (NP-complete) stochastic optimal control problems [39] and to MDPs with initially unknown parameters, for which RL heuristics may provide practical approximate solutions (Table 1). This heterogeneity in decision optimization problem types and achievable results suggests that no single type of causal explanation for decision recommendations is best for all situations. Rather, rationales for decision recommendations should be matched to the problem type.

## 9. Discussion: Explaining CAI Decision Recommendations

The different causal models we have reviewed lead to different detailed algorithms for decision optimization, but all draw on a small set of causal artificial intelligence (CAI) principles for making and defending (by explaining their rationales in causal terms) decision and policy recommendations when actions have uncertain consequences. This section attempts to distil those principles. In this context, a *decision* is a choice from a choice set of feasible alternatives. A *policy* is a decision rule specifying how to make a decision whenever one must be made, based on the information available then. A fully specified probabilistic causal model gives the probabilities of outcomes of actions. In influence diagrams (IDs), these are probabilities for the utility node, mediated by changes in the distributions of other variables caused by decisions. In Markov decision processes (MDPs), semi-Markov decision processes (SMDPs), and other stochastic optimal control models, actions can affect probabilistic state transition rates and reward distributions for occupying and/or transitioning between states. In partially observable MDPs (POMDPs) or partially observable SMDPs (POSMDPs), actions may also affect the information channels that map the unobserved underlying states to conditional probabilities of observed signals. In all of these classes of causal models, *causally effective decision-making* recommends decisions and policies that make preferred outcomes more probable, where preferences are represented by utility functions, reward functions (including discount rates for future rewards), or goal states (perhaps with forbidden states and safety constraints), depending on the model. A *causal explanation* for a recommended decision or policy explains *why* it is recommended, i.e., why it is believed that it will make preferred outcomes more probable.

CAI principles and computational methods that have proved useful for supporting causally effective decision-making and control in a variety of applications include the following.

**Expected utility (EU) optimization with known causal models:**
*Choose a feasible action or policy to maximize expected utility.* This is practical when probabilistic causal models are known and optimization is tractable. Decision problems with known causal models represented by small (tabular) Bayesian networks or influence diagrams (IDs), decision trees, or response surfaces (Figure 8 and Figure 9) are amenable to EU optimization-based decision recommendations and causal explanations, with the fundamental rationale for recommended decisions being that they are implied by the normative axioms of EU theory.**Dynamic decision optimization and risk management with known dynamic causal models:**
*Make current decisions to maximize a long-run objective function*, taking into account potential future decisions and chance events. Common objective functions include long-run average, total, or discounted rewards, and long-run expected utility. Dynamic optimization is practical for known causal models represented by small decision trees, influence diagrams based on dynamic Bayesian networks (DBNs), or MDPs. It is practical for some POMDPs, but computationally difficult for others, and impossible (undecidable) for some infinite-horizon POMDPs. MDP and POMDP solution algorithms can be modified to incorporate risk aversion, using a variety of coherent risk measures [63,65]. When dynamic optimization is impracticable or impossible, heuristics must be used to guide choice.**Causal simulation andMonte Carlo Tree Search (MCTS) heuristics**: *Make current decisions based on the best plan available when decisions must be made.* MCTS and other causal simulation methods use causal models to simulate possible futures and incrementally improve plans, i.e., sequences of decisions contingent on future events, with increasing CPU time. MCTS does so by selecting (via adaptive sampling), simulating, and evaluating possible futures and keeping the best (most highly valued) plans discovered. Although full optimization of an objective function may be impossible with the time and resources provided, causal simulation uses the best plan discovered so far to recommend what to do when decisions must be made. It is practical if a probabilistic causal model, e.g., a POMDP or a discrete-event simulation model, is available to support its simulations (“roll outs”) of possible futures.**Reinforcement Learning (RL) with initially unknown causal models:**
*Make current decisions to reduce expected regret, taking into account the estimated value of information from actions*, as well as the estimated immediate state transitions and rewards that the actions cause. RL with entropy regularization is useful for discovering optimal or near-optimal (low-regret) policies in MDPs, even when relevant causal models linking actions to probabilities of consequences (immediate rewards and state transitions) are initially unknown [74].**Information and control capacities:**
*Make current decisions based on realistic assessments of what is possible to accomplish in future.* If an AI makes or recommends decisions in order to control a system that evolves dynamically in response to a mix of controlled and uncontrolled (stochastic) inputs, then the set of probability distribution over outcomes that it can reach and maintain in a given time interval via its control signals is limited by its control capacity—the rate at which it can transfer information to future states of the system—as well as by its capacity to sense and process relevant information needed for causally effective control. To the extent that these constraints are understood, or can be learned from experience, they can inform planning by showing what goals (construed as desired probability distributions over states reached within specified times) are realistically achievable. While this principle has been most thoroughly developed for discrete-time linear dynamical systems, the link between incoming and outgoing information rates and causally effective controllability of systems is applicable more generally [82] and is being further clarified via recent work (e.g., [83]).

These principles are complementary. Current state-of-the-art AI/ML systems for discovering how to act effectively under uncertainty typically integrate several of them. For example, RL techniques such as Thompson sampling are now commonly used in MCTS to prioritize scenarios to simulate and evaluate. RL, MCTS, and numerical methods for approximate stochastic dynamic programming and policy optimization have all benefitted from the use of flexible nonparametric function approximators, such as deep neural networks, to estimate value functions for evaluating policies from limited samples of decision and outcome trajectories. AI/ML systems that integrate dynamic decision optimization, RL, and MCTS methods have proved useful in a wide range of applications, both when complex causal models (e.g., POMDPs) are available, and also in the absence of an *a priori* known causal model (model-free RL).

### Applying CAI Principles to Explain Decision Recommendations

A powerful motivation for CAI is its potential to identify decisions and policies that improve average rewards (or improve other, risk-adjusted, objective function values) for decision and control under uncertainty, compared to methods that do not use causal models to help optimize decisions. We have emphasized an additional motivation: the potential for CAI to explain the causal rationales for its decision recommendations, thereby increasing their transparency and trustworthiness. However, the CAI methods we have discussed—EU maximization, dynamic optimizationwith or without risk aversion or safety constraints, and MCTS or optimal control, if causal models are known; RL if causal models are initially uncertain or unknown; and hybrids of these methods, assisted by function approximators, such as deep learning—are best understood by specialists. To what extent, then, can CAI facilitate humanly satisfying explanations that succeed in making decision recommendations more transparent, trusted, and acceptable for non-specialists? Although this question can only be fully answered empirically via social science research informed by the psychology of explanations [4,84], the distinctions we have discussed among model-based optimization, causal simulation heuristics, experiential learning (RL), and capacity to effect change may be useful in setting expectations for the different kinds of information that can, should, and must be included to create convincing explanations for decision and policy recommendations.

Current thinking on “explainable AI” (XAI) focuses largely on explanations for AI/ML black-box *predictions*, with palatable explanations generated by simplifying complex black-box predictive models to yield simpler (e.g., linear or threshold), approximate predictive rules [4]. By contrast, any rational explanation for a *decision* recommendation must include information about each of the following components:*A causal model describing how outcome probabilities depend on choices*. Examples of causal models discussed in previous sections include conditional probability tables (CPTs) in influence diagrams and in dynamic Bayesian networks (DBNs); response surface models; and MDPs, POMDPs, stochastic control models, and causal simulation models for dynamic systems. Examples of outcomes over which preferences are defined in these various causal models include trajectories (time sequences) of system states, outputs, or rewards; terminal states to which utilities are attached; and realizations of the utility random variable in an influence diagram model. If a unique causal model is not known, then model ensemble methods (e.g., different estimated causal models produced by different ML techniques, as in the slightly different PDP contour maps in Figure 9) or reinforcement learning methods that maintain posterior distributions over possible models (e.g., beta distributions for the rewards from each arm of a MAB, or from each act in each state of a MDP, in Thompson sampling) may be used to reflect model uncertainty [85].*Preferences for outcomes.* Preferences are represented in several ways in the different causal models we have discussed, e.g., by a utility function; by an objective function for trajectories of states and outputs, in POMDPs and stochastic control models, perhaps modified or constrained to reflect desired risk aversion; by estimated one-step transition rewards for state transitions and/or occupancy, together with a discount rate, in MDPs; or by goal states to be reached and taboo states to be avoided over some time horizon in stochastic optimal control models.*A choice set of feasible alternatives that the decision-maker can currently choose among.* While POMDPs, behavior trees, and control capacity models all acknowledge that a decision-maker may not be able to simply choose an action that is then implemented promptly and without error, all of them also recognize that decisions involve selecting from a feasible set of choices—even if what is selected is only which control signal to send or what task(s) to attempt next, rather than the successful completion of an action.

Given these three main components, a CAI decision support algorithm applies optimization methods or heuristics to recommend decisions that are predicted by the causal model to make preferred outcomes more probable. If a causal model is known for which optimization is tractable, then CAI recommends decisions to maximize an objective function. For example, Figure 9 illustrates how a choice set (the budget line), possible outcomes (sales levels), a causal model learned from data (the iso-sales contours expressed as functions of the decision variables, namely, advertising in each channel), and assumed preferences (higher contours are assumed to be preferred) come together in a simple traditional nonlinear programming optimization model. If optimization is intractable, then causal simulation heuristics, such as MCTS, are used. If no causal model is available, then reinforcement learning is used.

In each case, the optimization algorithms or heuristics for selecting a recommended decision are likely to be inscrutable to many users. Their results are not likely to be readily explicable except in simple cases, such as that shown in Figure 9. However, the three required inputs (causal model, preference model, choice set) can often be explained and visualized with the help of Bayesian networks and influence diagrams (or decision trees and tables, for very small problems); state transition diagrams and example trajectories for MDPs and POMDPs; and response surface models, partial dependence plots (PDPs), and accumulated local effects (ALE) plots, if there are only a few continuous decision variables. Moreover, if a trusted causal simulation model is available, then the recommendations from a CAI-based decision recommendation engine can be made credible, even if the optimization algorithms that lead to them are obscure, by simulating deviations from them and showing that non-recommended policies reduce the value of the objective function being maximized (or stochastically reduce it, i.e., shift its cumulative distribution leftward over repeated simulation runs). If a trusted causal model is not available, then sensitivity analyses and reinforcement learning methods that maintain and update posterior distributions for the best policies can be used instead to show how deviating from recommended decisions reduces expected utility, as assessed with currently available information.

From this perspective, explaining CAI-based decision recommendations to build trust and acceptance among users need not confront the challenge of explaining how the CAI algorithms work. Rather, they can focus on assuring that the choice sets, preference models, and causal models that the optimization algorithms or heuristics use to generate recommendations are well understood and agreed to, and that the resulting decision recommendations are credible, as shown by the demonstrable difficulty (or, for exact optimization methods, impossibility) of improving upon them. For decades, human decision analysts have emphasized precisely these features in communicating with and earning the trust of decision-makers: it is essential to be clear about the inputs and outputs and the structure of the analysis (choice sets, possible outcomes, preferences and value tradeoffs among outcomes, and conditional probabilities for outcomes given choices), but technical details for solution algorithms can be safely encapsulated in influence diagram software or other solvers with the validity and robustness of results (i.e., decision recommendations) being clarified more through simulations and sensitivity analyses than by explaining how they were calculated [86,87]. To this traditional perspective from decision analysis on how to build trust in analyses and recommendations, we add the following points:a.In recommending a decision or policy, a CAI advisory system should disclose whether its recommendation is based on full optimization, heuristic search, or RL exploration.b.It should also disclose whether its recommendation is based on a well-validated causal model (e.g., a causal model with CPTs for which properties of transportability and invariant causal prediction (ICP) have been tested and verified); on a causal model that has been assumed but not yet validated (as in many traditional decision analyses that use subjectively elicited decision trees or IDs); on an uncertain model or ensemble of plausible causal models learned from data (e.g., with contours in Figure 9 estimated using causal PDP or ALE plots based on ensemble ML techniques); or on reinforcement learning (RL) trajectories seeking to estimate from experience how decisions affect rewards or risk-adjusted objective functions.c.If the relevant causal model is uncertain (e.g., if it is still being learned by an RL heuristic that has not yet converged to a zero-regret answer), then the CAI systems should disclose current uncertainty about the best recommendation. It should note how, if at all, risk aversion and safe exploration constraints have been incorporated into the search for the best recommendation.d.If heuristics have been used to make a decision recommendation, then bounds on regret, or other performance guarantees, should be provided if possible. For example, such information is produced by several current state-of-the-art POMDP heuristics [53,78].

While explaining details of CAI algorithms is probably unnecessary for justifying decision recommendations and explaining to what extent they are trustworthy, explaining the technical basis for the analyses that supports the recommendations—especially the above aspects of causal model uncertainty and optimization uncertainty—can help to set realistic expectations for the reliability of the recommendations.

## 10. Conclusions: Explaining Recommended Decisions in Causal AI

Needing to show that following a CAI advisory system’s decision recommendations is expected to cause probability distributions for possible outcomes that are preferred to the distributions caused by feasible alternatives creates a distinctive set of information requirements and a distinctive structure for causal explanations of decision recommendations. These differ markedly from the information requirements and structure needed to explain observations or predictions. Explaining decision rationales requires not simply conditional probability distributions, but causal models of effects of possible (counterfactual) actions on probability distributions. It requires being able to use knowledge of choice sets, preferences (e.g., utility functions or goal states) for possible future outcomes, and estimated causal links (e.g., CPTs satisfying ICP) between current actions and probabilities of future outcomes to explain why the recommended course of action is best, as well as procedural knowledge of appropriate exact or heuristic optimization methods for finding the recommended course of action. In relatively simple situations, decision recommendations are then obtained by optimizing a known (or approximately known) objective function that serves as a causal model for predicting the value or expected utility of outcomes caused by different feasible settings of decision variables. Figure 9 illustrates this concept. Feasible allocations of the available budget between marketing channels constitute the choice set; estimated iso-sales contours constitute the causal model; and sales correspond to the outcome variable to be maximized. The recommended decision (the starred point of tangency between the budget constraint line and the estimated contours of expected sales caused by different actions, i.e., allocations) has a clear explanation in this example: deviating from the recommendation in either direction causes lower performance contours.

Such a causal model also readily explains how and why recommended decisions change as conditions change, even if the new conditions have never been encountered before. For example, a change in the budget or in advertising prices (changing the budget line) or in advertising effectiveness (changing the contours) that implies a new point of tangency in the example of Figure 9 thereby immediately implies a new optimal budget allocation decision, even if no data have been collected for the new situation. In this way, causal models can provide defensible recommendations for new situations without requiring further training data, as long as the relevant causal laws are known, and are known to apply for the new conditions (transportability and ICP). This enables a CAI advisory system that monitors changes in causally relevant conditions (e.g., parents or ancestors of the utility node in an ID model) within the scope of known causal laws (causal CPTs) to exercise *automated vigilance* on behalf of those it advises, proactively recommending next optimal actions in response to observed changes in conditions without having to wait for new training data or to learn new predictive patterns. The recommendations can be explained by showing how estimated optimal decisions change as conditions change, similar to comparative statics in economics. For more elaborate dynamic causal simulation models, explanations cannot necessarily be visualized as easily as in Figure 9, but simulating the consequences of deviating from recommended policies can provide convincing evidence that the recommendations should be followed, provided that the causal simulation model is trusted. To validate a causal model, in turn, an array of tools from causal analysis is available, as discussed in the section on the structure of explanations in causal Bayesian networks. These include structure-learning algorithms, CPT estimation algorithms, and conditional independence tests for implications of hypothesized causal structures; statistical tests for internal validity based on comparison of causal effect estimates across multiple distinct adjustment sets; and statistical tests for external validity based on transportability and invariant causal prediction (ICP) [2,30]. As with all statistical tests, passing these various validation tests does not assure that a causal model is valid, but provides evidence against the null hypothesis that its testable predictions are false. These methods have been most extensively developed in the context of causal Bayesian networks, but they apply also to other causal models, including MDPs and POMDPs, that can be represented by equivalent dynamic Bayesian networks [44].

If relevant causal laws (CPTs satisfying ICP) are known for only a limited range of conditions, however—perhaps only those in the vicinity of the currently estimated optimal policy, if it has been learned via RL—then changes that push observed conditions outside the scope of the currently known CPTs may require new rounds of RL before confident decision recommendations can be made. In Figure 9, for example, if the response surface had been fit to a narrow range of data near the starred optimum instead of to a relatively wide data cloud, the resulting causal PDP model might have produced a relatively narrow swath of well-estimated contours surrounded by unknown contours outside the range of observed data. To characterize the reliability of the basis for its recommended decisions, CAI advisory systems should, therefore, not only distinguish among exact optimization, heuristic (sub)-optimization, and exploratory RL heuristics, but they should also characterize remaining uncertainty about their recommendations whenever possible, e.g., by providing regret bounds or performance guarantees. Such uncertainty information differs from the statistical confidence intervals and Bayesian posterior distributions typically used in predictive analytics, insofar as it requires counterfactual comparisons—for example, for the expected difference between achieved rewards and the maximum rewards that would have been achieved had the truly optimal policy been known and followed. Plausible explanations for why one action should be taken instead of another make such counterfactual comparisons necessary. Causal models of the effects of actions on the outcome probabilities make them possible.

Although CAI is recent enough to make confident prediction of trends difficult, it has recently been identified as an important next step for the evolution of applied AI/ML by AI/ML developers, users and commentators (e.g., [17,88,89]). The main reason is that it allows AI to go beyond pattern-driven inferences and predictions to calculate the probable effects of different interventions, or courses of action, and to optimize policies and behaviors accordingly—in effect, leading to smarter AI/ML. As explained by [89]: “Ultimately, knowing the ‘why’ behind complex problems helps us to understand how the world really operates and, in turn, to identify the right actions to achieve desired outcomes. We may yet find that an ounce of causal AI is worth a pound of prediction.” For practitioners, the greatest practical value of CAI is probably that it helps to identify actions and policies that are causally effective in increasing the probabilities of achieving desired outcomes [12]. For developers, a strong contribution of CAI is that it improves the speed and accuracy of machine learning, especially in novel situations, by enabling efficient generalization and transfer of knowledge learned under previously encountered conditions to apply to new situations, as in the example of Figure 9. CPTs satisfying the ICP property need not be re-learned every time an environment changes, or as new conditions or interventions are encountered, but instead can be used to “transport” previously acquired causal knowledge to the new situations (via transportability calculations) to predict effects of interventions without waiting to collect new data [26].

In some applications areas, such as Systems Biology, users and developers have collaborated closely in developing and applying CAI [11]. CAI implementation and public-domain software have advanced enormously in the past decade, and uniform interfaces to multiple causal discovery packages are now readily available, such as the *CompareCausalNetworks* package in R [17]. Despite these advances, the situation illustrated in Figure 5 and Figure 8 (left panel), in which the structure of a causal model is not uniquely determined automatically from data alone, is still common in practice. As a result, current CAI technology is often used as a tool by knowledgeable experts to generate and test causal hypotheses and quantify causal relationships. Human experts apply common-sense causal knowledge (such as that age and sex can be causes, but not effects, of diseases; or that death can be an effect, but not a cause, of diseases and exposures) to constrain the search for causal models that are consistent with the data [11,18]. Whether used as a tool by human experts, or used by autonomous or human-guided AI to speed learning and to help optimize decisions, CAI will most likely be trusted in the future if it can give clear explanations for its conclusions. We have proposed that, for decisions—as opposed to predictions, diagnoses, or causal discovery—the structure of convincing explanations involves the components shown in Figure 1 for a wide range of causal models (Table 1); making this structure clear can provide a useful and credible explanation for CAI-based decision recommendations, even if details of policy optimization heuristics are not discussed. Future systems that incorporate causal explanations together with CAI methods may help to increase the trustworthiness and acceptability of CAI in practical applications for proposing improved decisions and policies under uncertain and changing conditions.

We have discussed the structures and types of information and arguments needed to give convincing causal explanations for recommended decisions or policies. The essence of such an explanation is a counterfactual comparison between the changes in outcome probabilities that the recommended decisions or policies are expected to cause, on the one hand, and those that alternative decisions would cause, on the other. This is precisely the type of comparison that optimization and RL heuristics use to make their recommendations. Therefore, the same CAI methods can be used both to identify *what* decisions to recommend, as well as to explain *why* they are recommended. Emphasizing causally explainable decision recommendations improves their quality as well as their credibility. This is because, in contrast to simplification-based XAI for predictive analytics, CAI for prescriptive analytics emphasizes the production of evidence that following the recommendations makes desired outcomes more probable. We propose that demonstrable effectiveness in trusted causal simulation models, rather than simplicity, is the essence of convincing explanations for decision recommendations from CAI-assisted prescriptive analytics.

## Data Availability

Data sources are listed in the R code for the figures.

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
