# Peer review of "Information Structures for Causally Explainable Decisions"

_entropy, 2021, doi:10.3390/e23050601_

Round 1
Reviewer 1 Report
The paper has reviewed explainable AI concepts explaining decision recommendations. The review focuses on casual models of the relationship between actions and outcome probabilities. In general, the paper has very potential as a journal paper.
The authors can consider the following questions and suggestions to improve the quality of the paper:
- It would be better if the introduction section can provide a diagram that can outline the paper content and summarize all the concepts, principles and methods reviewed in this paper.
- To make the introduction better and support the introduction presentation of CAI principles, the paper can draw a diagram that indicates the position of CAI in the research area of explainable AI.
- To help readers get a better understanding of CAI in the introduction section, the paper can provide one practical application example of CAI and describe how the use of CAI improves the trustworthy.
- I am not sure about the purpose of section 2 (Methods and results: information for statistical vs. causal explanations) in the paper. Please revise section 2 carefully!
- It is not clear about the methodology which the paper used to construct the review. Additional methodology section can express how the paper collects and selects data for the review.
- Some references are not presented properly to provide essential information such as Ahmadi, 2021; Crowley, 2004; Du, 2020, etc. Please check the reference section! All the references need to be presented in the correct format and enough information.
- Figure 4 and figure 7 (left) are not informative. What is the meaning of arrows’ colours?
- In general, there are AI developers and AI users. AI developers (e.g. data scientists, software engineering) are expert in AI while AI users (e.g. field workers, domain experts) has limited knowledge and understanding of AI. AI users only use AI as a tool to help them decision-making in their domain application. In those cases, how CAI principles and methods help AI developers and users.
- I am curious about the review of CAI implementation. What are the advantages and disadvantages of CAI implementation for particular projects regarding available hardware and software?
- The paper can make some predictions of the future of CAI and its development trend in the conclusion section.
Author Response
Please see the attachment. Thank you for your helpful suggestions! I have revised the paper to respond to them, as detailed in the attachment (alsopasted below, but without the formatting).
The paper has reviewed explainable AI concepts explaining decision recommendations. The review focuses on casual models of the relationship between actions and outcome probabilities. In general, the paper has very potential as a journal paper.
The authors can consider the following questions and suggestions to improve the quality of the paper:
RESPONSE: Thanks so much for these constructive suggestions.
- It would be better if the introduction section can provide a diagram that can outline the paper content and summarize all the concepts, principles and methods reviewed in this paper.
- To make the introduction better and support the introduction presentation of CAI principles, the paper can draw a diagram that indicates the position of CAI in the research area of explainable AI.
RESPONSE: Great suggestions, thanks! Rather than two diagrams, I added a diagram (the new Figure 1; all other figures are renumbered) and a table, which was more compact for the amount of information to be summarized. Then new text is as follows:
“Figure 1 summarizes key concepts and methods reviewed in subsequent sections and shows how they fit together. Observations (upper left) provide information about the underlying state (lower left) of the system or environment via an information channel, i.e., a probabilistic mapping from the state to observations. Actions (lower right) cause state transitions and associated costs or benefits, generically referred to as rewards (bottom) via a causal model, i.e., a probabilistic mapping from current state-action pairs to conditional probabilities of next-state and reward pairs. Table 1 list several specific classes of causal models discussed later, in rough order of increasing generality and flexibility in representing uncertainty. Actions are selected by policies, also called decision rules, strategies, or control laws (upper right), which, at their most general, are probabilistic mappings from observations to control signals sent to actuators; these control signals are the decision-maker’s or controller’s choices. The mapping from choices to actions may be probabilistic if actuators are not entirely reliable; in this case, the capacity of the control channel and actuators to transfer information from control signals to future states of the system limits the possibilities for control. In this framework, observations are explained by underlying states (and the information channels via which they generate observations). By contrast, rational decisions are explained via optimization of decision-rules (i.e., policies). If knowledge of the causal model and the other components in Figure 1 is inadequate to support full optimization, then reinforcement learning or other adaptive control methods, together with optimization heuristics, are used to improve policies over time. The vast majority of “explainable AI” (XAI) research to date has focused on explaining observations (e.g., diagnostic systems), predictions, and prediction-driven recommendations (e.g., Mittelstadt et al., 2019), emphasizing the left side of Figure 1, where observations are used to draw inferences about states that help to predict observations based on conditional expected values. A principal goal of this paper is to help extend XAI to more fully explain the rationales for recommended decisions and policies, using preferences for outcomes (i.e., rewards), choice sets (e.g., possible control signals), causal models, and optimization of policies as key explanatory constructs, in addition to observations and inferences.”
Figure 1. Summary of key ideas and methods used in causal AI (CAI) to explain decisions
policy
optimization
observations ® policy/decision rule, P(a | y)
information channel, P(y | x) ¯ control signal
current state, x action
¯ ¯ causal model: P(x’, r|x, a) = P(next state, reward | current state, action),
outcome: probabilistic transition to next state; probabilistic reward
Symbols :x = current state, x’ = next state, y = observation, a = action, r = reward
Table 1. Some important probabilistic causal models
|
Probabilistic causal models, in order of increasing generality |
Knowledge representation, inference, and decision optimization assumptions |
|
Decision trees (Raiffa, 1968), event trees (decision trees without decisions); fault trees, event trees, bow tie diagrams (Cox et al., 2018) |
Event trees show possible sequences of events (realizations of random variables). Decision trees are event trees augmented with choice nodes and utilities at the leaves of the tree. Fault tree are trees of binary logical events and deterministic logic gates, supporting bottom-up inference from low-level events to top-level event (e.g., systems failure). Bow-tie diagrams integrate fault trees leading up to an event and event trees following it. |
|
Bayesian networks (BNs), dynamic BNs, causal BNs
Influence diagrams (IDs) are BNs with decision nodes and utility nodes (Howard and Matheson, 1981) |
Random variables (nodes) are linked by probabilistic dependencies (described by conditional probability tables, CPTs). In a DBN, variables can change over time. In a causal BN, changing a variable changes the probability distributions of its children in a directed acyclic graph (DAG). Bayesian inference of unobserved quantities from observed (or assumed) ones can proceed in any direction. |
|
Markov decision process (MDP) optimization models, can be risk-sensitive |
Markov transition assumptions, observed states, actions completed without delays |
|
Partially observable MDPS (POMDPs) |
States are not directly observed, but must be inferred from observations (signals, symptoms, data) via information channels, P(observation = y | state = x) |
|
PO semi-MDPs (POSMDPs); behavior trees |
Actions take random amounts of time to complete, and may fail |
|
Discrete-event simulation models |
Realistic lags and dependencies among events are modeled by state-dependent conditional intensities for individual-level transitions |
|
Causal simulation-optimization models |
Known models. Inference and optimization can be NP-hard and may require heuristics such as Monte Carlo Tree Search (MCTS). |
|
Model ensemble optimization; reinforcement learning (RL) with initially unknown or uncertain causal models |
Unknown/uncertain models. Probabilistic causal relationships between actions and consequences (e.g., rewards and state transitions) are learned via (heuristic-guided) trial and error. |
- To help readers get a better understanding of CAI in the introduction section, the paper can provide one practical application example of CAI and describe how the use of CAI improves the trustworthy.
RESPONSE: Good idea. I added the following discussion to the Introduction:
“Our focus is on CAI concepts, principles and methods for causal explanation of decisions, but a variety of practical applications have been described in industrial engineering and industrial control, managerial economics (e.g., for forestry or fishery management), personalized medicine, supply chain management, logistics optimization, urban traffic control, robotics, autonomous vehicle and drone control, pest management in ecosystems, financial investments, game-playing, and other areas of applied risk analysis (e.g., Cox et al., 2018). Many of these applications have focused solely on decision optimization, rather than also on decision explanation, under risk and uncertainty. This motivates our focus on the structure of causal explanations under risk and uncertainty. However, to understand how CAI methods and explanations can improve the trustworthiness of causal inferences and intervention decision recommendations in practice, we recommend recent analyses and applications of computational causal methods in Systems Biology for cancer research. Although current CAI methods do not yet fully automate valid causal discovery with high reliability (Triantafillou et al., 2017), causal discovery and understanding of low-level (molecular-biological) pathways are increasingly able to inform, and build confidence in, high-level public health policies by helping target the right causal factors at the macro-level (e.g., diet, exposures) to be causally effective in reducing risks (Vineis et al., 2017). Explaining how interventions cause desired changes helps select effective interventions.”
- I am not sure about the purpose of section 2 (Methods and results: information for statistical vs. causal explanations) in the paper. Please revise section 2 carefully!
RESPONSE: Thanks – I agree. I have eliminated that section heading, and folded the text into the remaining paper. (I originally thought that Methods and Results headings were required by the journal, but on closer reading, I see they are optional.)
- It is not clear about the methodology which the paper used to construct the review. Additional methodology section can express how the paper collects and selects data for the review.
RESPONSE: I added Figure 2 and Table 1 to clarify the conceptual outline for what is covered. In addition, I added the following text:
“2. Methodology and Applications
To accomplish this review, synthesis, and exposition of CAI for explaining decisions, we first consider the traditional concept of explanation in statistics as the proportion of variance in a dependent variable in a regression model that is “explained” by differences in the independent variables on which it depends. A different concept is needed to explain decisions, since regression coefficients only address how to predict dependent variables from observed independent variables, but not the causal question of how intervening to exogenously change independent variables would change probability distributions of dependent variables (Pearl, 2010). We therefore examine the structure of causal explanations for decisions in the causal models in Table 1. These models have been widely used in causal analytics, risk analysis, and applied AI/ML to represent probabilistic dependencies of outcomes (e.g., rewards and next states, in a Markov Decision Process) on actions (Cox et al., 2018). For each model, we discuss how the concepts in Figure 1 can be used to recommend decisions and explain their rationales. The discussion includes relatively recent developments (e.g., integration of Thompson sampling into Monte Carlo Tree Search) while noting foundational works in prescriptive decision analysis and decision optimization (e.g., Bellman, 1957; Howard, 1960; Howard and Matheson, 1981). ”
- Some references are not presented properly to provide essential information such as Ahmadi, 2021; Crowley, 2004; Du, 2020, etc. Please check the reference section! All the references need to be presented in the correct format and enough information.
RESPONSE: If the paper is accepted, I will indeed work closely with the journal’s copy-editor to assure that all references are presented in the correct format using the most recent available information as of the date of the final version. However, please note that these references are currently available only on-line. Specifically, Ahmadi et al. 2021 says not to cite the AAAI version, stating that “The AAAI Digital Library will contain the published version some time after the conference.” Accordingly, I cited, the arXiv preprint repository version URL (https://arxiv.org/abs/2012.02423) and the version at https://www.researchgate.net/publication/348664012_Constrained_Risk-Averse_Markov_Decision_Processes; I cited the arXiv version because it is stable (and citable),and I included the aaai link as a place-holder for the soon-to-be-published version. The published version should be citable very soon, probably in time for this paper. Similarly, for Du et al. (2020), only the arXiv version is currently available. The exact citation is the one I have given, per the citation instructions at https://www.semanticscholar.org/paper/When-is-Particle-Filtering-Efficient-for-POMDP-Du-Hu/aacce4c6b286d4c2c5e9fc8df9af0b8acdff6282. Likewise, the Crowley et al. 2004 paper, although quite famous, is (to my knowledge) only available on-line, and has no better citation than the URL given. (Usually, in my experience, the papers in the arXiv repository are published fairly quickly after appearing there, and I will give full journal citations wherever possible as I finalize the references with Entropy, if the paper is accepted, including specifying last-access-date (if needed) for papers that are only available via arXiv or other on-line sources.)
- Figure 5 and figure 8 (left) are not informative. What is the meaning of arrows’ colours?
RESPONSE: Added the information or darker and lighter colors to each figure: “All 4 algorithms agree on the heavy arrows (darker color), but only 2 of 4 agree on each light arrow (lighter color).”
- In general, there are AI developers and AI users. AI developers (e.g. data scientists, software engineering) are expert in AI while AI users (e.g. field workers, domain experts) has limited knowledge and understanding of AI. AI users only use AI as a tool to help them decision-making in their domain application. In those cases, how CAI principles and methods help AI developers and users.
- I am curious about the review of CAI implementation. What are the advantages and disadvantages of CAI implementation for particular projects regarding available hardware and software?
- The paper can make some predictions of the future of CAI and its development trend in the conclusion section.
RESPONSE: In response to these three points, I have added the following new discussions and references to the Conclusions:
“Although CAI is recent enough to make confident prediction of trends difficult, it has recently been identified as an important next step for the evolution of applied AI/ML by AI/ML developers, users and commentators (e.g., Dhar, 2020; Heinze-Deml and Meinshausen, 2020; Sgaier et al., 2020). The main reason is that it allows AI to go beyond pattern-driven inferences and predictions to calculate the probable effects of different interventions, or courses of action, and to optimize policies and behaviors accordingly – in effect, leading to smarter AI/ML. As explained by Sgaier et al. (2020), “Ultimately, knowing the ‘why’ behind complex problems helps us to understand how the world really operates and, in turn, to identify the right actions to achieve desired outcomes. We may yet find that an ounce of causal AI is worth a pound of prediction.” For practitioners, the greatest practical value of CAI is probably that it helps to identify actions and policies that are causally effective in increasing the probabilities of achieving desired outcomes (Vineis et al., 2017). For developers, a valuable contribution of CAI is that it improves the speed and accuracy of machine learning, especially in novel situations, by enabling efficient generalization and transfer of knowledge learned under previously encountered conditions to apply to new situations, as in the example of Figure 9. CPTs satisfying the ICP property need not be re-learned every time an environment changes, or as new conditions or interventions are encountered, but instead can be used to “transport” previously acquired causal knowledge to the new situations (via transportability calculations) to predict effects of interventions without waiting to collect new data (Heinze-Deml et al., 2017).
In some applications areas, such as Systems Biology, users and developers have collaborated closely in developing and applying CAI (Triantafillou et al. 2017). CAI implementation and public-domain software have advanced enormously in the past decade, and uniform interfaces to multiple causal discovery packages are now readily available, such as the CompareCausalNetworks package in R (Heinze-Deml et al., 2020). Despite these advances, the situation illustrated in Figures 4 and 7 (left panel), in which the structure of a causal model is not uniquely determined automatically from data alone, is still common in practice. As a result, current CAI technology is often used as a tool by knowledgeable experts to generate and test causal hypotheses and quantify causal relationships. Human experts apply common-sense causal knowledge (such as that age and sex can be causes, but not effects, of diseases; or that death can be an effect, but not a cause, of diseases and exposures) to constrain the search for causal models that are consistent with data (Nagarajan et al., 2013; Triantafillou et al. 2017). Whether used as a tool by human experts, or used by autonomous or human-guided AI to speed learning and help optimize decisions, CAI will be most likely to be trusted in future if it can give clear explanations for its conclusions. We have proposed that for decisions – as opposed to predictions, diagnoses, or causal discovery – the structure of convincing explanations involves the components in Figure 2 for a wide range of causal models (Table 1); and making this structure clear can provide a useful and credible explanation for CAI-based decision recommendations, even if details of policy optimization heuristics are not discussed. Future systems that incorporate causal explanations together with CAI methods may help to increase the trustworthiness and acceptability of CAI in practical applications to proposing improved decisions and policies under uncertain and changing conditions.”

Reviewer 2 Report
The paper describes decision rationales and models for Causal AI using recommendations as an example application area. The author gives an overview on causal models and causal analysis for explaining decision.
The paper is interesting and relatively easy to read since it is well structured. Concepts for causal AI well described. The discussion section is very good and useful.
My only complains are related to language and formatting:
- The formatting of "R" codes should be improved with a fixed with font suitable for source code (lines 155–163, 190–192, 424–427, 680–682, 700–706)
- The Figure 6 has three line plots with three different styles. Is there a reason? Otherwise it would be more pleasant to use the same style for each plot.
- The paper should be proof-read regarding missing and extra white spaces.
- Line 1048: Missing bullet point?
Author Response
Thank you for your help. Responses are attached, and also pasted below.
Reviewer 2
The paper describes decision rationales and models for Causal AI using recommendations as an example application area. The author gives an overview on causal models and causal analysis for explaining decision.
The paper is interesting and relatively easy to read since it is well structured. Concepts for causal AI well described. The discussion section is very good and useful.
RESPONSE: Thank you! This was a fun paper to write, and I appreciate your comments and suggestions for polishing it.
My only complains are related to language and formatting:
- The formatting of "R" codes should be improved with a fixed with font suitable for source code (lines 155–163, 190–192, 424–427, 680–682, 700–706)
RESPONSE: Done, thanks!
- The Figure 7 has three line plots with three different styles. Is there a reason? Otherwise it would be more pleasant to use the same style for each plot.
RESPONSE: I agree that it would be more pleasing to have the same style, but the three plots are produced by three different packages with different graphics (pdp, ICEbox, and ALEPlot) and there is no easy way to change the graphics, as the packages are produced and published to the CRAN repository by different groups. I have added this information on the different packages used for Figure 7, as follows: “Plots are produced using the pdp, ALEPlot, and ICEbox packages in R, respectively, by the Causal Analytics Toolkit at http://cloudcat.cox-associates.com:8899/.” (Note: A new Fogure 1 was added, and all other figures have been renumbered accordingly.)
- The paper should be proof-read regarding missing and extra white spaces.
RESPONSE: Yes, thanks. I am afraid this reflects a known MS Word compatibility bug. I have down-saved the source file to .doc instead of .docx to avoid this bug, and have fixed some of the mis-spacings in the current version, but will have to scrub the final (typeset) proofs of the whole paper carefully (if it is accepted), as this bug will almost certainly keep injecting spacing errors as long as the document is being transferred in MS Word. (For more on this problem, see https://support.microsoft.com/en-us/topic/spaces-are-missing-between-some-words-when-you-open-a-word-2010-document-in-word-2007-87478acf-5da7-37ce-1af3-5cdbdc5b9782. The serice pack to fix it is no longer available, so I think careful manual proofing of the page proofs once they are in .pdf is the only fix.)
- Line 1048: Missing bullet point?
RESPONSE: Fixed, thank you!
